# Complex fitness landscape shapes variation in a hyperpolymorphic species

Anastasia V Stolyarova[1]*, Tatiana V Neretina[2,3], Elena A Zvyagina[2,4], Anna V Fedotova[1,3], Alexey S Kondrashov[5], Georgii A Bazykin[1,6]*

[1]Skolkovo Institute of Science and Technology, Moscow, Russian Federation; [2]White Sea Biological Station, Biological Faculty, Lomonosov Moscow State University, Moscow, Russian Federation; [3]Belozersky Institute of Physico-Chemical Biology, Lomonosov Moscow State University, Moscow, Russian Federation; [4]Yugra State University, Khanty-Mansiysk, Russian Federation; [5]Department of Ecology and Evolutionary Biology, University of Michigan, Ann Arbor, United States; [6]Institute for Information Transmission Problems of the Russian Academy of Science, Moscow, Russian Federation

*For correspondence:
anastasia.v.stolyarova@gmail.
com (AVS);
gbazykin@iitp.ru (GAB)

Competing interest: The authors declare that no competing interests exist.

**Abstract** It is natural to assume that patterns of genetic variation in hyperpolymorphic species can reveal large-scale properties of the fitness landscape that are hard to detect by studying species with ordinary levels of genetic variation. Here, we study such patterns in a fungus *Schizophyllum commune*, the most polymorphic species known. Throughout the genome, short-range linkage disequilibrium (LD) caused by attraction of minor alleles is higher between pairs of nonsynonymous than of synonymous variants. This effect is especially pronounced for pairs of sites that are located within the same gene, especially if a large fraction of the gene is covered by haploblocks, genome segments where the gene pool consists of two highly divergent haplotypes, which is a signature of balancing selection. Haploblocks are usually shorter than 1000 nucleotides, and collectively cover about 10% of the *S. commune* genome. LD tends to be substantially higher for pairs of nonsynonymous variants encoding amino acids that interact within the protein. There is a substantial correlation between LDs at the same pairs of nonsynonymous mutations in the USA and the Russian populations. These patterns indicate that selection in *S. commune* involves positive epistasis due to compensatory interactions between nonsynonymous alleles. When less polymorphic species are studied, analogous patterns can be detected only through interspecific comparisons.

## Editor's evaluation

This study investigates a highly polymorphic species, the fungus Schizophyllum commune, and finds that, compared to synonymous mutations, levels of linkage disequilibrium between nonsynonymous mutations are higher within genes than between genes. The authors propose this observation may be explained by compensatory interactions between nonsynonymous alleles, pointing to the presence of positive epistasis. These exciting results provide insights into what levels of polymorphism can lead to the emergence of positive epistasis. This paper should be of interest to population geneticists and evolutionary biologists studying the role of natural selection.

## Introduction

Alleles do not affect fitness and other phenotypic traits independently and, instead, engage in epistatic interactions (*de Visser et al., 2011*; *de Visser and Krug, 2014*; *Gillespie, 1994*; *Good and Desai, 2015*; *Kryazhimskiy et al., 2011*; *Maynard Smith, 1970*; *McCandlish et al., 2013*; *Povolotskaya*

**eLife digest** Changes to DNA known as mutations may alter how the proteins and other components of a cell work, and thus play an important role in allowing living things to evolve new traits and abilities over many generations. Whether a mutation is beneficial or harmful may differ depending on the genetic background of the individual – that is, depending on other mutations present in other positions within the same gene – due to a phenomenon called epistasis.

Epistasis is known to affect how various species accumulate differences in their DNA compared to each other over time. For example, a mutation that is rare in humans and known to cause disease may be widespread in other primates because its negative effect is canceled out by another mutation that is standard for these species but absent in humans. However, it remains unclear whether epistasis plays a significant part in shaping genetic differences between individuals of the same species.

A type of fungus known as *Schizophyllum commune* lives on rotting wood and is found across the world. It is one of the most genetically diverse species currently known, so there is a higher chance of pairs of compensatory mutations occurring and persisting for a long time in *S. commune* than in most other species, providing a unique opportunity to study epistasis.

Here, Stolyarova et al. studied two distinct populations of *S. commune*, one from the USA and one from Russia. The team found that – unlike in humans, flies and other less genetically diverse species – epistasis maintains combinations of mutations in *S. commune* that individually would be harmful to the fungus but together compensate for each other. For example, pairs of mutations affecting specific molecules known as amino acids – the building blocks of proteins – that physically interact with each other tended to be found together in the same individuals.

One potential downside of having pairs of compensatory mutations in the genome is that when the organism reproduces, the process of making sex cells may split up these pairs so that harmful mutations are inherited without their partner mutations. Thus, epistasis may have helped shape the way *S. commune* and other genetically diverse species have evolved.

*and Kondrashov, 2010*). Epistasis is pervasive at the scale of between-species differences, where it is saliently manifested by Dobzhansky-Muller incompatibilities and results in low fitness of inter-specific hybrids (*Callahan et al., 2011*; *Corbett-Detig et al., 2013*; *Dobzhansky, 1936*; *Kondrashov et al., 2002*; *Orr, 1995*; *Taverner et al., 2020*). By contrast, at the scale of within-population variation, the importance of epistasis remains controversial (*Crow, 2010*; *Hill et al., 2008*; *Hivert et al., 2021*). This may look like a paradox, because such variation provides an opportunity to detect epistasis through linkage disequilibrium (LD), non-random associations between alleles at different loci (*Beissinger et al., 2016*; *Boyrie et al., 2021*; *Garcia and Lohmueller, 2021*; *Wang et al., 2012*; *Zan et al., 2018*). In the case of positive epistasis, a situation when a combination of alleles confers higher fitness than that expected from selection acting on these alleles individually, it can maintain favorable coadapted combinations of alleles at interacting sites, increasing linkage disequilibrium (LD) between them (*Barton, 2010*; *Boyrie et al., 2021*; *Kouyos et al., 2007*; *Pedruzzi et al., 2018*; *Takahasi and Tajima, 2005*). In sexual populations, recombination competes with epistasis, disrupting such coupling LD (*Neher and Shraiman, 2009*; *Pedruzzi et al., 2018*). Nevertheless, within a single gene, physical proximity alone may suffice to limit recombination, so sets of coadapted variants may evolve (*Dobzhansky, 1950*; *Lewontin and Kojima, 1960*). Such positive within-gene epistasis has been proposed to affect variation in natural populations (*Arnold et al., 2020*; *Ragsdale, 2021*), but conditions for this are expected to be restrictive (*Hansen, 2013*; *Mäki-Tanila and Hill, 2014*; *Sackton and Hartl, 2016*).

Perhaps, the fitness landscape is complex macroscopically but is more smooth microscopically or, in other words, epistasis is genuinely more pronounced at a macroscopic scale (*Ochs and Desai, 2015*). If so, studying epistasis in hyperpolymorphic populations, where differences between genotypes can be as high as those between genomes of species from different genera or even families, holds a great promise because variation within such a population can cover multiple fitness peaks or a sizeable chunk of a curved ridge of high fitness (*Bateson, 1909*; *Dobzhansky, 1937*; *Gavrilets, 1997*; *Kondrashov et al., 2002*; *Muller, 1942*; Appendix 1).

## Results

### Elevated LD between nonsynonymous polymorphisms

In a vast majority of species, nucleotide diversity π, the evolutionary distance between a pair of randomly chosen genotypes, is, at selectively neutral sites, of the order of 0.001 (as in *Homo sapiens*) or 0.01 (as in *Drosophila melanogaster*) (*Leffler et al., 2012*). Still, a few hyperpolymorphic species with π>0.1 are known, of which the wood-decaying fungus *Schizophyllum commune* is the most extreme, where π=0.20 or 0.13 in the USA or the Russian populations, respectively (*Baranova et al., 2015*; *Appendix 3—figure 1*). The two populations of *S. commune* are highly divergent (dS between populations ≈ 0.34, $F_{st}$ = 0.58), but there is essentially no structure within either of them (*Appendix 3—figure 2*). We studied 34 haploid genotypes from the USA and 21 from Russia, each obtained by sequencing and de novo assembly of a haploid culture originated from a single haplospore. The use of haploid samples and de novo assembly of each sample ensures robust identification of haplotypes. We then compared the LD between nonsynonymous SNPs ($LD_{nonsyn}$) to that between synonymous SNPs ($LD_{syn}$).

In both *S. commune* populations, at sites with minor allele frequency (MAF) >0.05, $LD_{nonsyn}$ is much higher than $LD_{syn}$ at the same nucleotide distance (*Figure 1A*, *Figure 1—figure supplement 1A*). This excess of $LD_{nonsyn}$ is much stronger for pairs of SNPs located within the same gene, compared to pairs of SNPs from adjacent genes at the same distance. By contrast, the excess of $LD_{nonsyn}$ is independent of whether the two SNPs are located within the same or in different exons of a gene (*Figure 1—figure supplement 2*). In *S. commune*, the recombination rate is higher within exons (*Seplyarskiy et al., 2014*), which may affect the patterns of LD; however, this factor could only reduce within-gene LD, and in any case cannot explain the difference between $LD_{nonsyn}$ and $LD_{syn}$. For *S. commune*, the excess of $LD_{nonsyn}$ over LDsyn holds when we explicitly control for MAFs (*Figure 1—figure supplement 3*), indicating that differences in MAFs between synonymous and nonsynonymous polymorphisms cannot explain the excess of $LD_{nonsyn}$.

A much weaker excess of $LD_{nonsyn}$ over $LD_{syn}$ for MAF >0.05 is also observed in the less genetically diverse *D. melanogaster* population (*Figure 1B*). In the still less polymorphic human populations, $LD_{nonsyn}$ is indistinguishable from $LD_{syn}$ at the same distances (*Figure 1C*, *Figure 1—figure supplement 1B*).

The excess of $LD_{nonsyn}$ over $LD_{syn}$ corresponds to the attraction between minor nonsynonymous alleles. This attraction can only appear due to positive epistasis between such alleles - higher-than-expected fitness of their combinations (Appendix 2). Positive epistasis can be expected to cause stronger LD in more polymorphic populations (*Figure 1D–F*, Appendix 1) and must be more common for pairs of sites located within the same gene, which are more likely to interact with each other.

For rare SNPs with MAF <0.05 taken alone, $LD_{nonsyn}$ is similar or lower to $LD_{syn}$ for all three species, consistent with the effects of random drift, Hill-Robertson interference, and/or negative epistasis (*Figure 1—figure supplements 4–6*; Appendix 2). Decreased LD between negatively selected polymorphisms is expected due to Hill-Robertson interference between deleterious alleles (*Hill and Robertson, 1966*; *Roze and Barton, 2006*); this effect has been described previously for *H. sapiens* (*Garcia and Lohmueller, 2021*) and *D. melanogaster* (*Sandler et al., 2021*) and is observed in our simulations (*Appendix 2—figure 4*). In addition, $LD_{nonsyn}$ can be reduced by negative epistasis between deleterious alleles (*Garcia and Lohmueller, 2021*), similarly to the negative LD detected among loss-of-function polymorphisms in humans, flies and plants (*Sandler et al., 2021*; *Sohail et al., 2017*).

### Elevated LD between interacting sites

Natural selection acting on physically interacting amino acids that are located close to each other within the three-dimensional structure of a protein is characterized by strong epistasis which leads to their coevolution at the level of between-species differences (*Marks et al., 2011*; *Ovchinnikov et al., 2014*; *Sjodt et al., 2018*). Genome-wide elevated LD between amino acid sites within structural domains was recently demonstrated in human populations (*Ragsdale, 2021*). Extraordinary diversity of *S. commune* makes it possible to observe an analogous phenomenon at the level of individual genes in within-population variation.

To test this, we aligned *S. commune* proteins to the PDB database of protein structures. In the obtained set of 5188 genes with a good match to a protein with known structure, we identified pairs of codons in the *S. commune* genome encoding amino acid residues positioned near each other

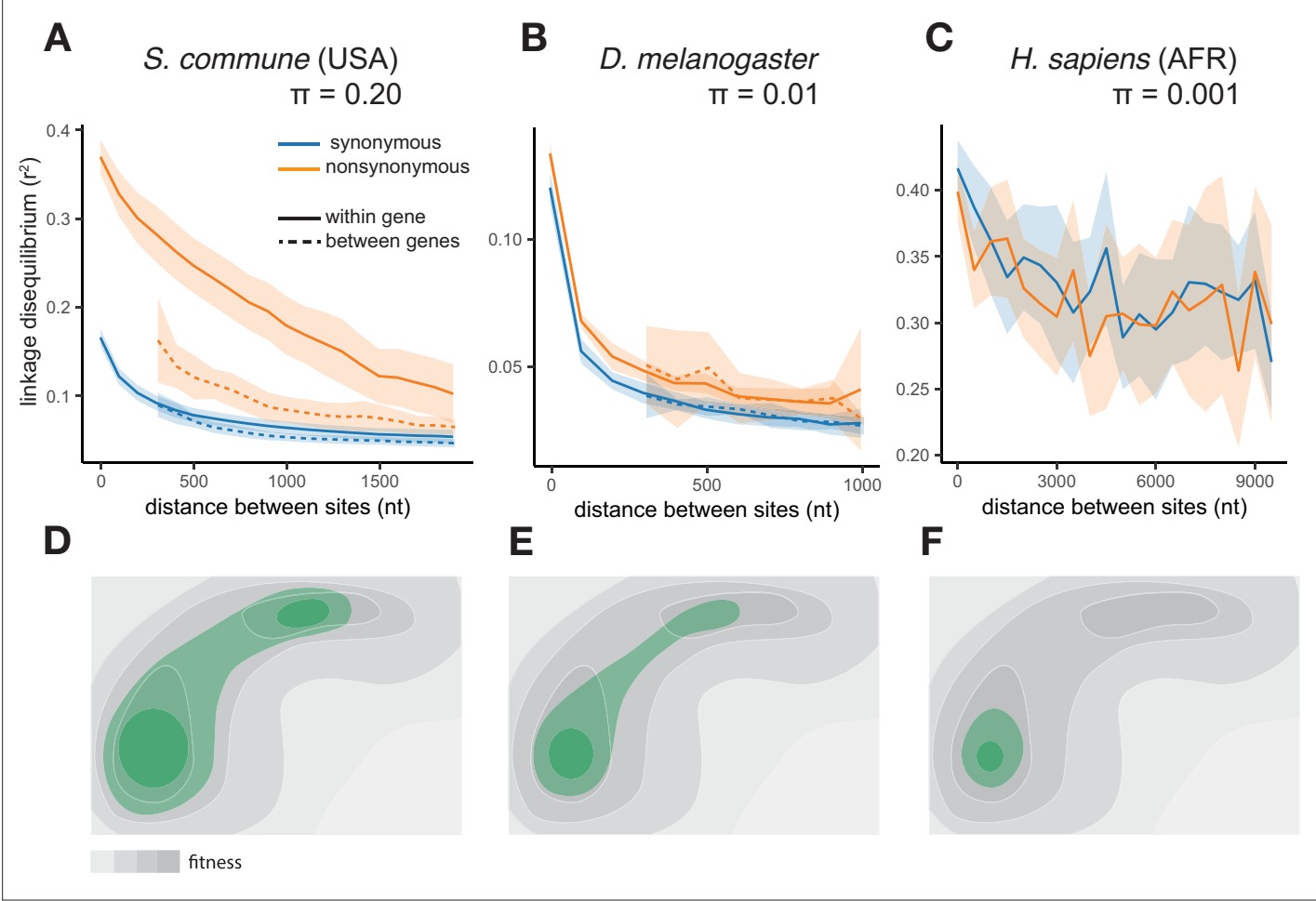

**Figure 1.** The efficiency of epistatic selection in populations with different levels of genetic diversity. (**A–C**) LD in natural populations for SNPs with MAF >0.05. (**A**) USA population of *S. commune*, (**B**) Zambian population of *D. melanogaster*, (**C**) African superpopulation of *H. sapiens*. Filled areas in (**A**)-(**C**) indicate SE of LD calculated for each chromosome or scaffold separately. (**D–F**) A hyperpolymorphic population (**D**) may occupy a sizeable chunk of a complex fitness landscape, leading to pervasive positive epistasis, while variation within less polymorphic populations (**E and F**) is confined to smaller, and approximately linear, portions of the landscape, so that no strong epistasis and LD can emerge. The area of the landscape covered by the population is shown in green.

The online version of this article includes the following figure supplement(s) for figure 1:

**Figure supplement 1.** The efficiency of epistatic selection in populations with different levels of genetic diversity.

**Figure supplement 2.** Linkage disequilibrium within and between exons in *S. commune*.

**Figure supplement 3.** Comparison of $LD_{nonsyn}$ and $LD_{syn}$ in *S. commune* populations with exact matching of both MAFs and distance.

**Figure supplement 4.** LD between SNPs with different MAF in *S. commune*.

**Figure supplement 5.** LD between SNPs with different MAF in *D. melanogaster*.

**Figure supplement 6.** LD between SNPs with different MAF in *H. sapiens*.

(within 10 Å) in the protein structures, and calculated the average LD between SNPs in such pairs of codons. Naturally, pairs of physically interacting sites are more likely to be closely spaced in the gene sequence and therefore to be under a higher LD than non-interacting ones. To account for this, we discarded pairs of SNPs within five amino acids from each other, and used a controlled permutation test (see Materials and methods) to compare the LD between physically close pairs of sites to that between distant pairs of sites.

In both *S. commune* populations, pairs of nonsynonymous SNPs are in stronger LD when they are located at codons encoding physically close than distant amino acids (*Figure 2A*; permutation test

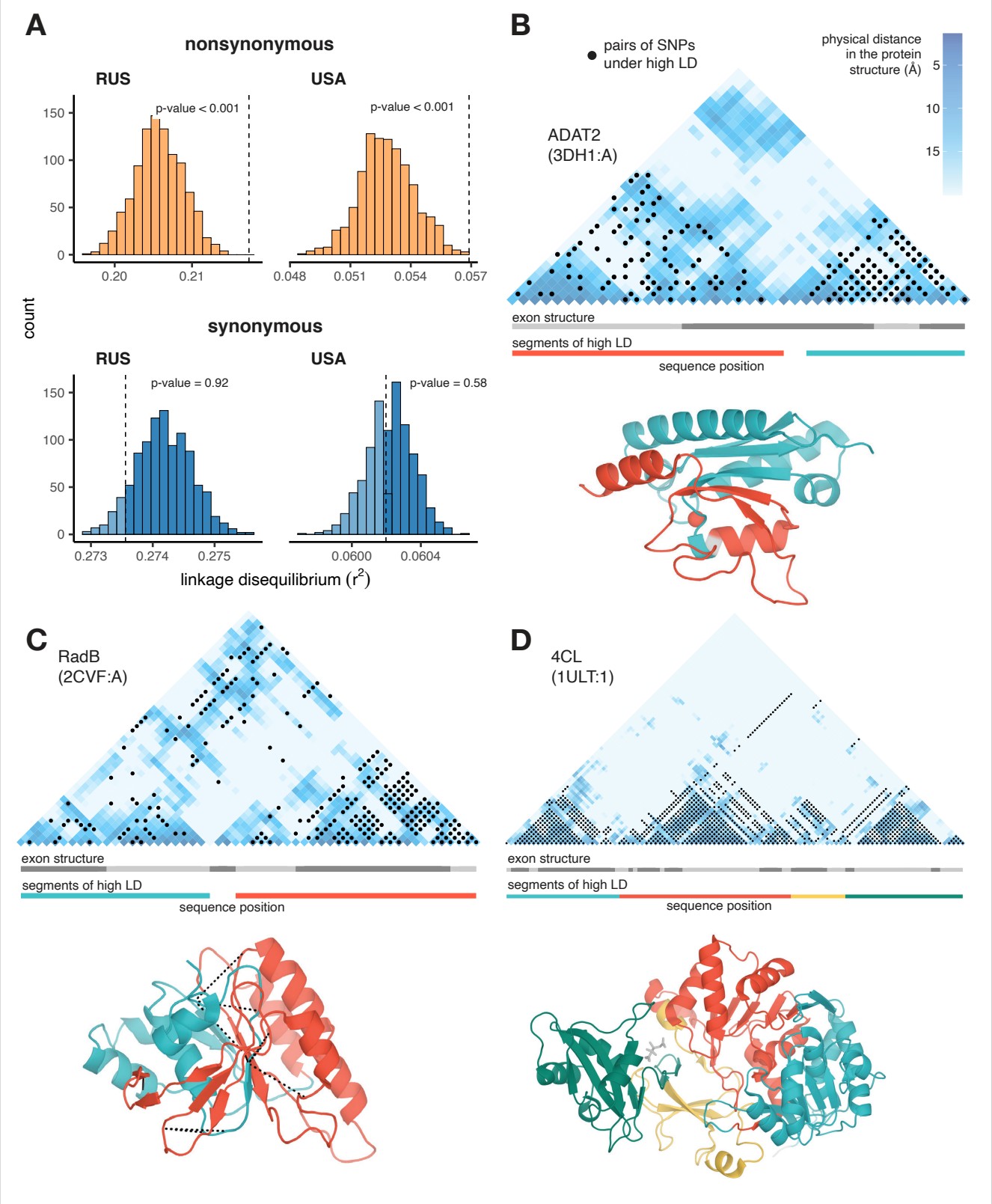

**Figure 2.** Excessive LD between physically interacting protein sites. (**A**) Within pairs of SNPs that correspond to pairs of amino acids that are colocalized within 10 Å in the protein structure, the LD is elevated between nonsynonymous, but not between synonymous, variants. Dashed lines show the average LD between colocalized sites. Permutations were performed by randomly sampling pairs of non-interacting SNPs while controlling for genetic distance between them, measured in amino acids; pairs of SNPs closer than 5 aa were excluded. (**B–D**) Examples of proteins with LD patterns matching their

*Figure 2 continued on next page*

*Figure 2 continued*

three-dimensional structures. Heatmaps show the physical distance between pairs of sites in the protein structure; only positions carrying biallelic SNPs are shown. Black dots correspond to pairs of sites with high LD (>0.9 quantile for the gene). Dashed lines in (**c**) structure show high LD between physically close SNPs from different segments of high LD. In these examples, LD is calculated in the Russian population of *S. commune*.

The online version of this article includes the following figure supplement(s) for figure 2:

**Figure supplement 1.** Examples of proteins with LD patterns matching the three-dimensional structure in the RUS population of *S. commune*.

**Figure supplement 2.** Examples of proteins with LD patterns matching the three-dimensional structure in the USA population of *S. commune*.

p-value <1e-3). This is not the case for pairs of synonymous SNPs (*Figure 2A*; permutation test p-value = 0.58).

Moreover, it is possible to identify individual proteins with significant associations between the patterns of LD and of physical interactions between sites. At a 5% FDR, we found 22 such proteins in the USA population, and 87 proteins in the Russian population (*Appendix 3—table 1*); three examples are shown in *Figure 2B–D* (see also *Figure 2—figure supplements 1 and 2*). The alignment of ADAT2 protein contains two segments (teal and red in *Figure 2B*) characterized by high within-segment LD. The boundaries of these segments match those of structural units of the protein, but not the exon structure of its gene. In RadB protein, a similar pattern is observed, and LD is also elevated between pairs of SNPs from different segments on the interface of the corresponding structural units (*Figure 2C*). The alignment of 4CL protein can be naturally split into four high-LD segments, which also match its structure (*Figure 2D*).

## Distinct regions of high LD

The magnitude of LD varies widely along the *S. commune* genome. Visual inspection of the data shows a salient pattern of regions of relatively low LD, alternating with mostly short regions of high LD (haploblocks, *Figure 3—figure supplement 1*). We calculated LD along the genome in a sliding window of 250 nucleotides and regarded as a haploblock any continuous genomic region with LD values that belong to the heavy tail of its distribution (see Materials and methods).

In the USA population, 8.4% of the genome is occupied by 5,316 such haploblocks, 56% consist of regions with background LD level, and the rest cannot be analyzed due to poor alignment quality or low SNP density. Eighty-eight percent of the haploblocks are shorter than 1000 nucleotides, although the longest haploblocks spread for several thousand nucleotides (*Figure 3—figure supplement 2*). In the Russian population, there are 10,694 haploblocks, occupying 15.9% of the genome, and regions of background LD cover 39% of it. There is only a modest correlation between the USA and Russian haploblocks: the probability that a genomic position belongs to a haploblock in both populations is 2.3% instead of the expected 1.3%, indicating their relatively short persistence time in the populations (examples shown in *Figure 3—figure supplement 1*).

LD within a haploblock is usually so high that most genotypes can be attributed to one of just two distinct haplotypes, which carry different sets of alleles (*Figure 3—figure supplement 3*). This results in a bimodal distribution of the fraction of minor alleles in a genotype within a haploblock, because some genotypes belong to the major haplotype and, thus, carry only a small fraction of minor alleles, and other genotypes belong to the minor haplotype and, thus, possess a high fraction of minor alleles (*Figure 3A*). Polymorphic sites within haploblocks are characterized by higher MAF than that at sites that reside in non-haploblock regions (t-test p-value <2e-16), and in the USA population MAFs within a haploblock are positively correlated with its strength of LD (*Figure 3B*, Pearson correlation estimate = 0.07, p-value <2e-6).

There is no one-to-one correspondence between haploblocks and genes, which are, on average, longer. Still, different genes are covered by haploblocks to different extent, which leads to wide variation in the strength of LD and other characteristics among them. Genes with high LD, that is those that contain haploblocks, have the largest excess of $LD_{nonsyn}$ over $LD_{syn}$ (*Figure 3C*). Positive correlation between the overall LD within the gene and the excess of $LD_{nonsyn}$ in this gene indicates that the attraction between nonsynonymous variants, driven by epistasis, is stronger if combinations of epistatic alleles are persisting within population for a long time, comprising haplotypes within a haploblock. Since both haplotypes tend to be common in a haploblock (*Figure 3*), this excess is much stronger for loci with MAF >0.05.

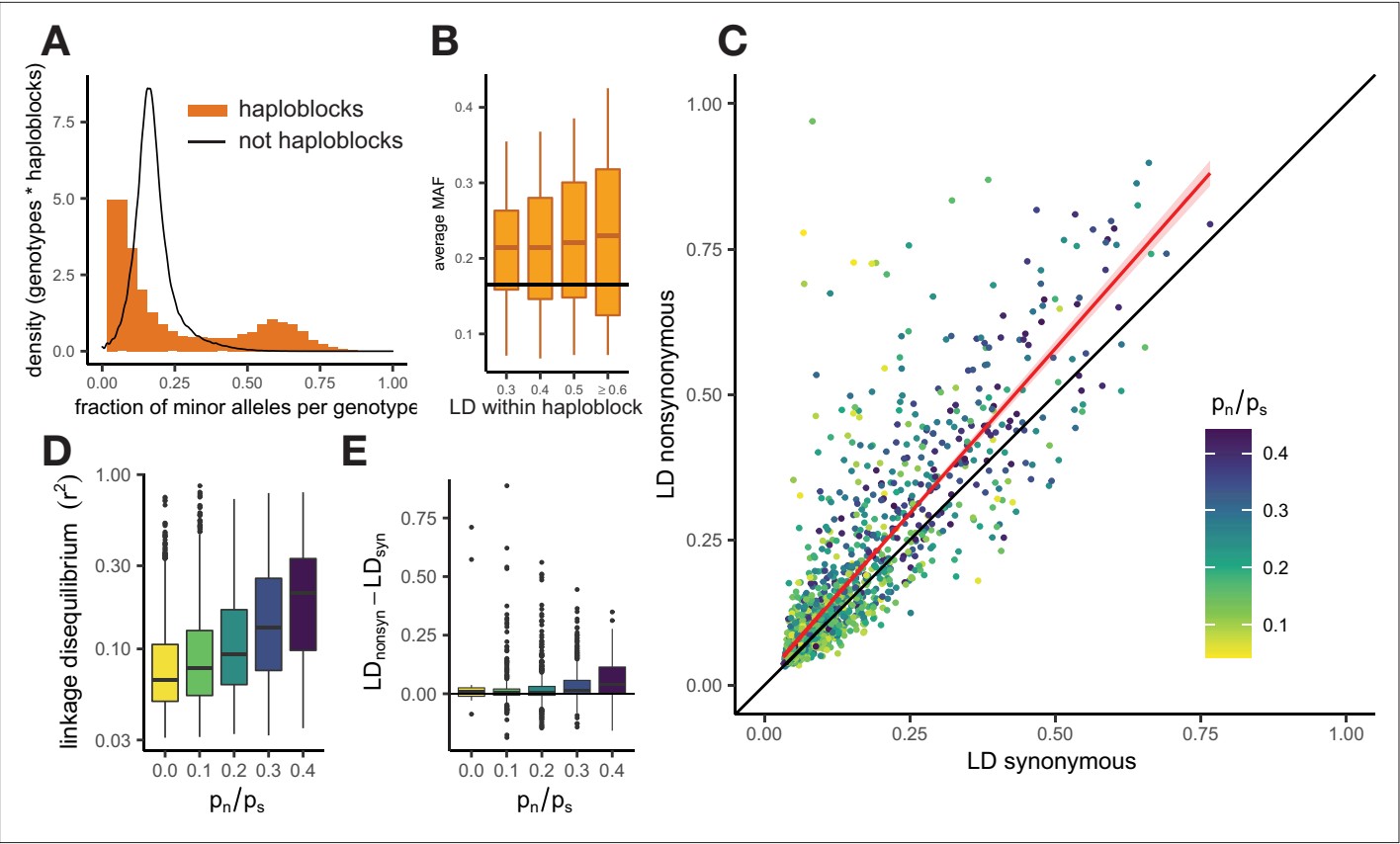

**Figure 3.** Patterns of linkage disequilibrium in the USA population of *S. commune*. (**A**) Distribution of the fraction of polymorphic sites that carry minor alleles in a genotype within haploblocks. Black line shows the distribution of fraction of minor alleles in genotypes in non-haploblock regions. (**B**) Distributions of the average MAF within a haploblock for haploblocks with different average values of LD. The average MAF in non-haploblock regions is shown as a horizontal black line for comparison. (**C**) LD between nonsynonymous and synonymous SNPs within individual genes. Linear regression of $LD_{nonsyn}$ on $LD_{nsyn}$ is shown as the red line. To control for the gene length, only SNPs within 300 nucleotides from each other were analyzed. Genes with fewer than 100 such pairs of SNPs were excluded. (**D,E**) The positive correlation between $p_n/p_s$ of the gene and its average LD (**D**) or the difference between $LD_{nonsyn}$ and $LD_{syn}$ (**E**). Here, the data on the USA population of *S. commune* are shown; similar patterns in the Russian population are shown in *Figure 3—figure supplement 4*.

The online version of this article includes the following figure supplement(s) for figure 3:

**Figure supplement 1.** Examples of haploblocks in two populations of *S. commune*.

**Figure supplement 2.** Distribution of haploblock lengths (nt) in the two populations of *S. commune*.

**Figure supplement 3.** Example of the *S. commune* alignment within a haploblock.

**Figure supplement 4.** Patterns of linkage disequilibrium in the RUS population of *S. commune*.

**Figure supplement 5.** Comparison of $LD_{nonsyn}$ and $LD_{syn}$ in the genes of *S. commune*.

**Figure supplement 6.** The difference between $LD_{nonsyn}$ and $LD_{syn}$ under pairwise epistasis and balancing selection.

**Figure supplement 7.** Criteria for haploblocks in *S. commune*.

LD between alleles of all kinds is higher within genes with large ratios of nonsynonymous and synonymous polymorphisms $p_n/p_s$ (Spearman correlation p-value <2e-16, *Figure 3D*). Genes with elevated $p_n/p_s$ also have a stronger excess of $LD_{nonsyn}$ over $LD_{syn}$ (*Figure 3E*, Spearman correlation p-value = 4.4e-17). This excess is the strongest for genes with high overall LD, but its correlation with $p_n/p_s$ holds even when the overall LD is controlled for (*Figure 3—figure supplement 5*).

There can be multiple non-exclusive mechanisms by which epistasis could lead to the observed positive associations between pn/ps, overall LD, and excess $LD_{nonsyn}$. First, genes under weaker selection, and therefore higher pn/ps, could be characterized by a higher overall amount and/or strength of epistasis. Second, epistasis, as estimated by excess $LD_{nonsyn}$, can contribute to increased pn/ps by allowing nonsynonymous polymorphisms to segregate in the population when maintained in

coadapted combinations, therefore weakening negative selection against them. Third, epistasis can be more potent in genes with lower overall recombination rate due to competition between epistasis and recombination: recombination breaks positively interacting combinations of alleles, disrupting linkage between them and interfering with epistasis. Fourth, existence of cosegregating combinations of mutually beneficial alleles could select for reduced local recombination rate.

## Excess of LD$_{nonsyn}$ requires stable polymorphism

Simulations show that positive epistasis alone cannot lead to the observed large excess LD$_{nonsyn}$ over LD$_{syn}$, for which two extra conditions need to be satisfied. The general reason for this is simple: in order for a substantial LD between not-too-rare alleles to appear, these alleles must persist in the population for a long enough time.

First, positive epistasis must lead to a full compensation of deleterious effects of individual alleles. In other words, the fitnesses of at least two most-fit genotypes that are present in the population at substantial frequencies must be (nearly) the same (***Figure 3—figure supplement 6***). If this is not the case, selection favoring the only most-fit genotype leads to a too low level of genetic variation, which persists only due to recurrent mutation. It is natural to assume that the two major haplotypes that are common within a haploblock correspond to high-fitness genotypes. High-fitness genotypes can represent either isolated fitness peaks of equal heights (corresponding to a situation when two out of the four allele combinations confer high fitness) or a flat, curved ridge of high fitness (corresponding to a situation when three out of four combinations confer high fitness). The available data are insufficient to distinguish between these two options. Of course, with complete selective neutrality of all allele combinations there is no reason for LD$_{nonsyn}$ >LD$_{syn}$, so that at least some mixed genotypes, carrying alleles from different high-fitness genotypes, must be maladapted.

Second, there must be some kind of balancing selection that specifically works to maintain variation, because otherwise random drift does not allow genetic variation to persist for a long enough time even if some, or even all, genotypes are equally fit (***Figure 3—figure supplement 6***). Here, there are at least two options. On the one hand, a 'real' negative frequency-dependent selection (NFDS) can act either directly at loci that display high LD or at some other tightly linked loci (***Charlesworth, 2006***; ***Olendorf et al., 2006***). On the other hand, variation can be maintained due to associative overdominance (AOD), resulting from selection against recurrent deleterious mutations at linked loci (***Gilbert et al., 2020***; ***Ohta, 1971***; ***Zhao and Charlesworth, 2016***).

Balancing selection is also neccessary for the presence of haploblocks, because a pair of divergent haplotypes can evolve in a panmictic population only if they coexist for a considerable time. A single locus under NFDS is enough to maintain a haploblock comprising the region of the genome around it. By contrast, if variation is maintained by AOD, it is more likely that selection against recessive mutations acts at a number of tightly linked loci (***Gilbert et al., 2020***). Long coexistence of diverged haplotypes that comprise a haploblock enables accumulation of co-adapted combinations of nonsynonymous alleles within them. Thus, it is not surprising that a pronounced excess of LD$_{nonsyn}$ over LD$_{syn}$ in *S. commune* is observed primarily within haploblocks and that this excess is higher in genes with higher p$_n$/p$_s$.

## Correlated LDs in two populations

Although a high excess of LD$_{nonsyn}$ is observed only within haploblocks, a signature of epistasis can also be seen outside of them in the form of a correlation between LDs in the two populations. This correlation can be high even if LDs by themselves are low.

The USA and the Russian populations share a large proportion of their SNPs. Given the high divergence between the two populations, few such shared SNPs are expected to have common origin in the ancestral population, and instead they are likely to have arisen from recurrent mutation. Since the haploblocks show little correlation between the two populations, we assume that they arose after their divergence. The high prevalence of coincident SNPs is not surprising because SNPs comprise 0.28 and 0.13 of all the aligned nucleotide sites in the USA and Russian populations, respectively (***Baranova et al., 2015***, ***Appendix 3—figure 2***). We identified pairs of shared biallelic SNPs located within 2 kb from one another and calculated the LD between them in both populations. To avoid the effects of strong within-population linkage and the occasional co-oc с urrence of haploblocks

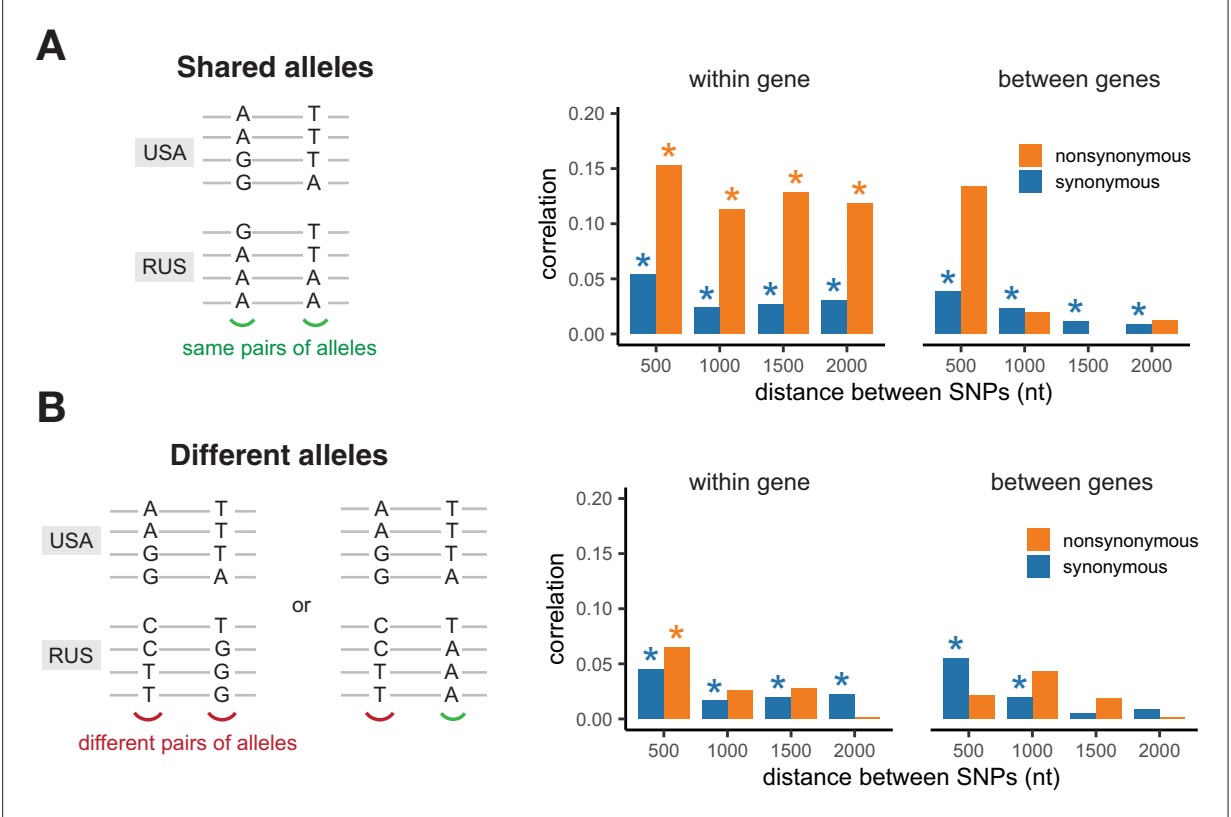

**Figure 4.** Correlation of LD values between pairs of shared SNPs in the two *S. commune* populations. (**A**) Pairs of SNPs with the same alleles in both sites, (**B**) pairs of SNPs differing by at least one allele. Asterisks indicate Spearman correlation p-values <0.001.

The online version of this article includes the following figure supplement(s) for figure 4:

**Figure supplement 1.** Association of LD values between pairs of shared nonsynonymous SNPs encoding the same amino acids in the two *S. commune* populations.

**Figure supplement 2.** Association of LD values between pairs of shared SNPs within haploblocks in the two *S. commune* populations.

between populations, we excluded SNPs located within haploblocks or within genes under high LD (>0.8 LD quantile for the corresponding population) in either population.

The values of LD in the two populations are strongly correlated only for pairs of nonsynonymous SNPs located within the same gene, and only if both populations carry the same pairs of amino acids in the same sites (*Figure 4*). The correlation of LDs is the strongest if shared SNPs carry the same pairs of nucleotides, but is also observed if they encode the same amino acids by different nucleotides (*Figure 4—figure supplement 1*). The contrast between correlations within pairs of sites that reside in the same vs. different genes and the correlation of LDs observed for different nucleotides encoding the same amino acid cannot be explained by inheritance of LD from the common ancestral population. Moreover, synonymous SNPs are expected to be on average older than nonsynonymous ones, so that this mechanism should lead to a higher correlation of LDs for pairs of synonymous mutations. Thus, the observed pattern indicates that epistatic selection is shared between the two populations.

The correlation of LDs between SNPs located within haploblocks in both populations is high regardless of whether they reside in the same or different genes, apparently because of occasional coincidence of haploblocks between populations (*Figure 4—figure supplement 2*).

## Discussion

On top of its most salient property, an exceptionally high π, genetic variation within *S. commune* possesses two other pervasive features. The first is a high prevalence of mostly short haploblocks, genome segments comprising two or occasionally three distinct haplotypes, which is a signature of

balancing selection. The overall fraction of the genome covered by haploblocks is ~10%, which is about an order of magnitude higher than the fraction covered by detectable signatures of balancing selection in genomes of other species (*DeGiorgio et al., 2014*; *Leffler et al., 2013*; *Rasmussen et al., 2014*).

The second feature is the excessive attraction between nonsynonymous alleles polarized by frequency. This pattern is much stronger within haploblocks, indicating that they were shaped by both balancing and epistatic selection, so that amino acids common within a haplotype together confer a higher fitness. Polymorphisms that involve haplotypes that comprise many interacting genes, such as inversions (*Charlesworth and Charlesworth, 1973*; *Dobzhansky and Pavlovsky, 1958*; *Singh, 2008*; *Sturtevant and Mather, 1938*) and supergenes (*Joron et al., 2011*; *Mather, 1950*), are known from the dawn of population genetics, but here we are dealing with an analogous phenomenon at a much finer scale, because haploblocks are typically shorter than genes. Thus, instead of coadapted gene complexes (*Dobzhansky and Pavlovsky, 1958*), haplotypes represent coadaptive site complexes within genes.

In our simulations, equally high fitnesses of two or more genotypes was a necessary condition for a large excess of $LD_{nonsyn}$, because otherwise the polymorphism did not live long enough for any substantial LD to evolve. However, epistasis between loci responsible for real or apparent balancing selection and those involved in compensatory interactions probably abolished the need for this fine-tuning of fitnesses. For example, if each haploblock carries its own complement of partially recessive deleterious mutations, together with alleles engaged in compensatory interactions with each other which also make these recessive mutations less deleterious, AOD can be expected to cause stable coexistence of these alleles.

Why are haploblocks and positive LD between minor nonsynonymous alleles so common in *S. commune*, but not in other, less polymorphic, species? There may be several, not mutually exclusive, reasons for this. Regarding haploblocks, real or apparent balancing selection may be more common in *S. commune* due to its higher polymorphism. Also, the same balancing selection may protect poly-morphism in a huge population of *S. commune*, but not in populations with lower $N_e$. Finally, an excess of haploblocks in *S. commune* may be at least due to better detection of signatures of balancing selection in a species with an extraordinary density of SNPs.

Excessive $LD_{nonsyn}$ in *S. commune* is also likely to be due to its hyperpolymorphism which increases the probability that mutually compensating alleles at a pair of interacting sites achieve high frequency and encounter each other in the same haplotype before being eliminated by selection. In other words, even if the fitness landscape remains the same, it results in more epistatic selection and, thus, in stronger LD in a species whose genetic variation covers a larger chunk of this landscape (*Figure 1*).

In a vast majority of species, π is a small parameter <<1. This imposes a severe constraint on oper-ation of selection and obscures signatures of its particular modes. Thus, hyperpolymorphic species where π is ~1 provide a unique opportunity to study phenomena which are traditionally viewed as belonging to the domain of macroevolution through data on within-population variation.

## Materials and methods
### *S. commune* sampling, sequencing, and assembly

Haploid cultures of 24 isolates, each originated from a single haplospore, were obtained from fruit bodies collected in Ann Arbor, MI, USA by T. James and A. Kondrashov and in Moscow and Kostroma regions, Russia by A. Kondrashov, A. Baykalova and T. Neretina in 2009–2015. Specimen vouchers are stored in the White Sea Branch of Zoological Museum of Moscow State University (WS). Herbarium numbers are listed in *Appendix 3—table 2*. To obtain isolates, wild fruit bodies were hung on the top lid of a 10 cm petri dish with agar medium. Petri dish was set at an angle of 60–70 degrees to the horizontal surface for 32 hr. A germinated spore was excised together with a square-shaped fragment (approximately 0.7 × 0.7 mm) of the medium from the maximally rarefied area of the obtained spore print under a stereomicroscope with 100 x magnification. The obtained isolates were cultured in Petri dishes on 2% malt extract agar for a week. For storage, cultures were subcultured into 1.5 ml micro-centrifuge tubes with 2% malt extract agar. To obtain sufficient biomass for DNA isolation, isolates

were cultured in 20 ml 0.5% malt extract liquid medium in 50 ml microcentrifuge tubes in a horizontal position on a shaker at 100 rpm in daylight for 5–10 days. The tubes with the cultures were then centrifuged at 4000 rpm, and the supernatant was decanted. The resulting mycelium was lyophilized. DNA was extracted using Diamond DNA kit according to the manufacturer's recommendations.

DNA libraries were constructed using the NEBNext Ultra II DNA Library Prep Kit kit by New England Biolabs (NEB) and the NEBNext Multiplex Oligos for Illumina (Index Primers Set 1) by NEB following the manufacturer's protocol. The samples were amplified using 10 cycles of PCR. The constructed libraries were sequenced on Illumina NextSeq500 with paired-end read length of 151. The genomes were assembled de novo using SPAdes (v3.6.0) (*Bankevich et al., 2012*); possible contaminations were removed using *blobology* (*Kumar et al., 2013*). Average N50 was ~165 kb for USA samples and ~70 kb for Russian samples (assembly statistics are provided in *Appendix 3—table 2*).

Together with the 30 samples sequenced previously (*Baranova et al., 2015*; *Bezmenova et al., 2020*), the obtained haploid genomes were aligned with TBA and *multiz* (*Blanchette et al., 2004*) and projected onto the reference scaffolds (*Ohm et al., 2010*). Ortholog sequences were extracted on the basis of the reference genome annotation (*Ohm et al., 2010*) and realigned using *macse* codon-based aligner (*Ranwez et al., 2011*). The alignments are available at https://makarich.fbb.msu.ru/astolyarova/schizophyllum_data/. Only the gap-free columns of the whole-genome alignment and the orthologs that were found in all 55 genomes were used for analysis. The total number of detected SNPs was 5.8 million for the USA population (82% of them biallelic) and 2.7 million for the Russian population (93% biallelic). 25% of the USA SNPs were shared with the Russian population (11% with the same major and minor alleles), and 53% of the Russian SNPs were shared with the USA population (23% with the same major and minor alleles, *Appendix 3—figure 1*).

The phylogeny of the sequenced genomes was reconstructed with RAxML (*Stamatakis, 2014*; *Appendix 3—figure 2*). Nucleotide diversity (π) was estimated as the average frequency of pairwise nucleotide differences; π for different classes of sites is shown in *Appendix 3—figure 1B*. Two samples from Florida (USA population) were excluded from the further analysis to minimize the possible effect of population structure.

Genome sequence data are deposited at DDBJ/ENA/GenBank under accession numbers JAGVRL000000000-JAGVSI000000000, BioProject PRJNA720428. Sequencing data are deposited at SRA with accession numbers SRR14467839-SRR14467862.

## Data on *H. sapiens* and *D. melanogaster* populations

We used polymorphism data from 347 phased diploid human genomes from African and 301 genomes from European super-populations sequenced as part of the 1000 Genomes project (*1,000 1000 Genomes Project Consortium et al., 2015*). If several individuals from the same family were sequenced, we included only one of them. As a *D. melanogaster* dataset, we used 197 haploid genomes from the Zambia population (*Lack et al., 2015*). Only autosomes were analyzed in both datasets.

## Estimation of LD

As a measure of linkage disequilibrium between two biallelic sites, we used r², calculated as follows: $r^2 = \frac{(p(AB) - p(A)p(B))^2}{p(A)(1-p(A))\ p(B)(1-p(B))}$, where p(A) and p(B) are the minor allele frequencies at these sites, and p(AB) is the frequency of the genotype that carries minor alleles at both sites.

Singletons (sites with minor allele present only in one genotype) were excluded from the analysis if not stated otherwise.

## Haploblocks annotation

In order to annotate the haploblocks, we calculated LD along the *S. commune* genome in a sliding window of 250 nucleotides with a step of 20 nucleotides (only non-singleton SNPs are analyzed; the windows with less than 10 SNPs were excluded). Any continuous sequence of overlapping windows with r² larger than the threshold value was merged together in a haploblock. The LD threshold value was defined independently for each *S. commune* population as the heavy tail of the within-window LD

distribution, as compared with the lognormal distribution with the same mean and variance as in the data (*Figure 3—figure supplement 7*).

## Estimation of LD between physically interacting amino acid sites

Of 16,319 annotated protein-coding genes of *S. commune* (*Ohm et al., 2010*) 9,941 were found in all 55 aligned genomes. We blasted the protein sequences of these orthologous groups against the PDB database of protein structures. About 52% of them (5,188) had a match (e-value threshold = 1e-5) amongst the proteins with the known structure. We realigned the sequences of *S. commune* protein and the matching PDB protein with *clustal* and calculated within-population LD and physical distance (Å) for each pair of aligned positions. A pair of amino acid sites was considered physically adjacent if they were located within 10 Å from each other.

To compare LD between pairs of physically close and distant sites, we used the controlled permutation test (*Figure 2A*): for each pair of physically close amino acid sites (within 10 Å) we sampled a pair of physically distant amino acids on the same genetic distance (measured in aa). Pairs of sites closer than 5 aa were excluded from the analysis.

To examine LD patterns within individual protein structures, we calculated contingency tables of pairs of SNPs being located in codons encoding physically close amino acids and having high LD (no less than 90% quantile for a given gene). Pairs of amino acid sites located closer than 30 aa or more distant than 100 aa from each other were excluded; genes with less than five pairs of physically close sites under high or low LD were also excluded. From these contingency tables, we calculated the odds ratio (OR) and chi-square test p-value for each gene. p-values were adjusted using BH correction. We identified 22 genes with pairs of adjacent sites having significantly higher LD in the USA population (out of 1286 eligible genes in total), and 87 genes in the Russian population (out of 967) under 5% FDR (*Appendix 3—table 1*). Examples of such genes are shown in *Figure 2* and *Figure 2—figure supplements 1 and 2*.

## Simulations of epistasis

To simulate evolution of populations with or without epistasis and balancing selection (*Figure 3—figure supplement 6*), we used an individual-based model implemented by *SLiM* (*Haller and Messer, 2019*). Simulations were performed with diploid population size N=1000 and recombination rate 0. To achieve the level of genetic diversity π similar to *S. commune*, mutation rate μ was scaled as μ=π/2N=5e-5. The length of the simulated sequence was 100 nt. Each simulation started with a monomorphic population and proceeds for 100 N generations. For calculations of synonymous and nonsynonymous LD, random 100 haploid genotypes were sampled from the population. Only SNPs with minor allele frequency >5% in the sample were analyzed.

We modelled two types of mutations, depending on whether they are neutral (with selection coefficient $s_{syn} = 0$) or weakly deleterious ($s_{nonsyn} \leq 0$), representing synonymous and nonsynonymous variants correspondingly. There were twice as many nonsynonymous as synonymous sites. Under the nonepistatic model, *s* was independent of the genetic background. We assumed $s_{nonsyn} = -0.01$ with the dominance coefficient *h* of 0.5.

Under the pairwise positive epistasis model, we assumed that one nonsynonymous mutation can be partially or fully compensated by a mutation at another site. In this model, all nonsynonymous sites were split into pairs. Each mutation of a pair individually occurring within a genotype was assumed to be deleterious, with selection coefficient $s_{nonsyn} = -0.01$; however, the fitness of the double mutant is larger than expected under the additive (non-epistatic) model. We used several models of epistasis, with different strengths of epistasis strength and landscape shapes (*Figure 3—figure supplement 6*).

In the NFDS model of balancing selection, a single mutation at a random position was subjected to frequency-dependent selection (so that it is positively selected at frequencies below 0.5, and negatively selected at frequencies above 0.5). In the AOD model, mutations in 10 random positions were fully recessive (h=0) and weakly deleterious (s=−0.0025).

To simulate evolution of populations with different levels of genetic diversity under epistasis (Appendix 1), we used *FFPopSim* (*Zanini and Neher, 2012*). To achieve different levels of genetic diversity π, mutation rate μ was scaled as μ=π/2N. The calculations were performed the same way as in *SLiM*, but In this case, we used haploid population size N=2000, population-scaled recombination rate 0.01 and the simulated sequence length of 300 nucleotides.

## Acknowledgements

We thank Timothy James, Anna Baykalova and members of Bazykin and Kondrashov labs for collecting *S. commune* samples. We thank Shamil Sunyaev and members of his lab for useful comments on drafts of this article.

## Additional information

### Funding

| Funder | Grant reference number | Author |
|---|---|---|
| Russian Science Foundation | 14-50-00150 | Georgii A Bazykin |

The funders had no role in study design, data collection and interpretation, or the decision to submit the work for publication.

### Author contributions

Anastasia V Stolyarova, Formal analysis, Investigation, Methodology, Visualization, Writing - original draft; Tatiana V Neretina, Investigation, Resources; Elena A Zvyagina, Anna V Fedotova, Investigation; Alexey S Kondrashov, Georgii A Bazykin, Conceptualization, Supervision, Writing – review and editing

### Author ORCIDs

Anastasia V Stolyarova http://orcid.org/0000-0002-6546-3052
Georgii A Bazykin http://orcid.org/0000-0003-2334-2751

### Decision letter and Author response

Decision letter https://doi.org/10.7554/eLife.76073.sa1
Author response https://doi.org/10.7554/eLife.76073.sa2

## Additional files

### Supplementary files

• MDAR checklist

### Data availability

Whole-genome alignment of 55 genomes of *S. commune* is available at https://makarich.fbb.msu.ru/astolyarova/schizophyllum_data/. Genome sequence data are deposited at DDBJ/ENA/GenBank under accession numbers JAGVRL000000000-JAGVSI000000000, BioProject PRJNA720428. Sequencing data are deposited at SRA with accession numbers SRR14467839-SRR14467862.

The following dataset was generated:

| Author(s) | Year | Dataset title | Dataset URL | Database and Identifier |
|---|---|---|---|---|
| Stolyarova AV, Neretina TV, Zvyagina AE, Fedotova AV, Kondrashov AS, Bazykin GA | 2021 | Genome sequencing and assembly | https://www.ncbi.nlm.nih.gov/bioproject/PRJNA720428 | NCBI BioProject, PRJNA720428 |

The following previously published datasets were used:

| Author(s) | Year | Dataset title | Dataset URL | Database and Identifier |
|---|---|---|---|---|
| The 1000 Genomes Project Consortium | 2015 | 1000 genomes project | https://www.internationalgenome.org/ | 1000 Genomes phase 3 release, 1000 |
| Lack JB, Cardeno CM, Crepeau MW, Taylor W, Corbett-Detig RB, Stevens KA, Langley CH, Pool JE | 2015 | Zambia population of *D. melanogaster* | http://johnpool.net/genomes.html | Drosophila Genome Nexus, DPGP3 |

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

## Appendix 1

### Epistatic selection is more efficient in genetically diverse populations

Genetic interactions affect operation of selection only in a sufficiently variable population. The potency of any kind of selection increases with the amount of variation; for epistatic selection, however, this increase is expected to be faster than linear, because it depends on the number of possible allele combinations. In a highly polymorphic population, a particular allele is more likely to co-occur in the same haplotype with an interacting, for example compensatory, allele (*Appendix 1—figure 1A*) which should increase the impact of epistasis on linkage disequilibrium.

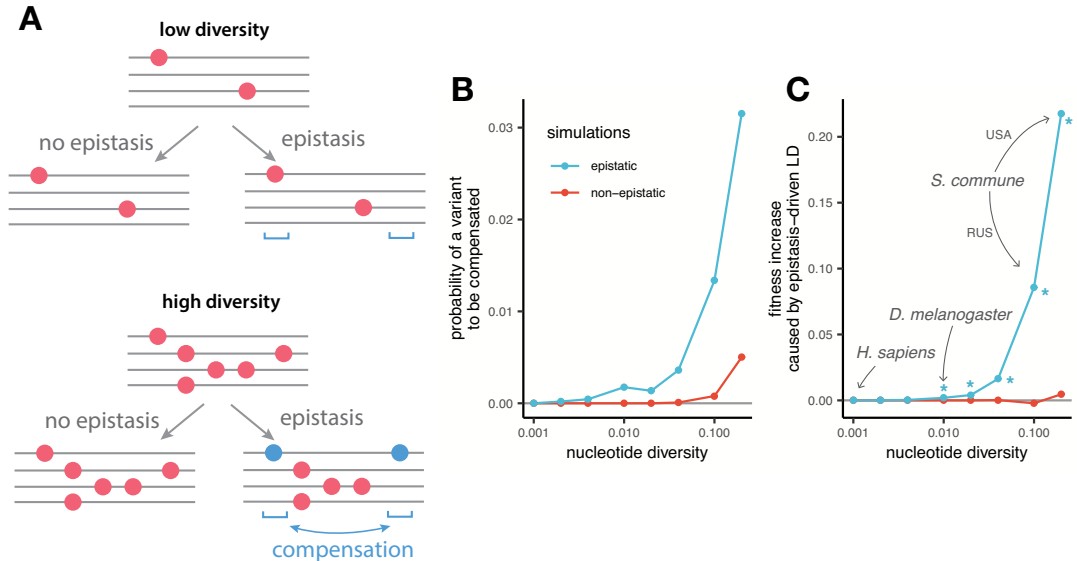

**Appendix 1—figure 1.** The efficiency of epistasis in populations with different levels of nucleotide diversity. (**A**) Under low nucleotide diversity, deleterious mutations (red dots) are unlikely to be compensated. If nucleotide diversity is high, epistatic selection maintains LD between SNPs in interacting sites (blue dots). (**B**) The probability that a deleterious variant is compensated by another variant within the same individual at the end of the simulation. (**C**) Increase in the mean fitness of the population caused by epistatic selection maintaining LD between favorable allele combinations. The fitness is plotted relative to that of a population consisting of individuals with uncorrelated alleles at different sites, obtained by permuting alleles among individuals. The efficiency of epistatic selection in maintaining linkage is much higher in genetically variable populations. Asterisks in (**C**) indicate significant deviation from 0 (Wilcoxon paired test p-value <0.01). Each simulation was repeated between 100 and 10,000 times depending on genetic diversity.

To illustrate this point, we modelled the evolution of a genome region in the presence and in the absence of positive epistasis in a panmictic population. We assumed that all mutations at a set of sites are individually deleterious, and that all these sites are involved in pairwise positive (i.e. antagonistic) sign epistasis; specifically, each deleterious mutation can be fully compensated by another mutation at exactly one site elsewhere in the genome, which is also deleterious when present alone. In the non-epistatic simulations, the effects of mutations were independent; however, at the end of the simulation we randomly assigned the 'interacting' pairs of sites to account for the random coincidence of deleterious alleles. We found that in this model a higher polymorphism increases the probability that a deleterious mutation is compensated before being eliminated by selection (*Appendix 1—figure 1B*). This probability increases with genetic diversity even for the non-epistatic simulations, because increased diversity elevates the likelihood of randomly encountering a compensating allele in the same haplotype. For epistatic simulations, however, this increase is more radical, reflecting the effect of epistatic selection favoring compensated haplotypes.

After the mutation-selection equilibrium was reached, we measured the strength of epistatic selection between all segregating polymorphisms, asking to what extent the mutational load is reduced by epistasis maintaining combinations of compensatory mutations. As shown in *Appendix 1—figure 1C*, the ability of epistatic selection to reduce the mutation load (i.e. to increase

the mean fitness) strongly depends on π. In less variable populations (π<0.01), epistasis is practically inefficient and does not affect LD (Wilcoxon test -values >0.33); this is because the probability of occurrence of the favorable combination of alleles in the population for selection to act upon is low. In more diverse populations, however, such combinations may arise and be favored by epistatic selection, which increases LD between them (Wilcoxon test p-value <0.01 for π≥0.01).

## Appendix 2

### LD$_{nonsyn}$ >LD$_{syn}$ requires positive epistasis

When alleles at different loci can be related to each other, it makes sense to consider the sign of both epistasis and LD. For example, if we are concerned with allele frequencies, all rare alleles can be viewed as analogous. Then, LD is positive (negative) if genotypes carrying both rare alleles are over-(under-)represented in the population, and epistasis is positive (negative) if such genotypes have fitnesses above (below) those expected if alleles act independently. Of course, overrepresentation (and excessive fitness) of combinations of rare alleles automatically entails the same for combinations of common alleles.

Although we report LD between pairs of polymorphic sites as r², which is symmetric regarding the major or minor variants, the observed high values of r² correspond to positive LD between minor alleles for both synonymous and nonsynonymous SNPs (*Appendix 2—figures 1–3*). Thus, LD$_{nonsyn}$ >LD$_{syn}$ means that attraction between minor nonsynonymous alleles is stronger than between minor synonymous alleles. This pattern may seem to be surprising, because there are three factors that work in the opposite direction.

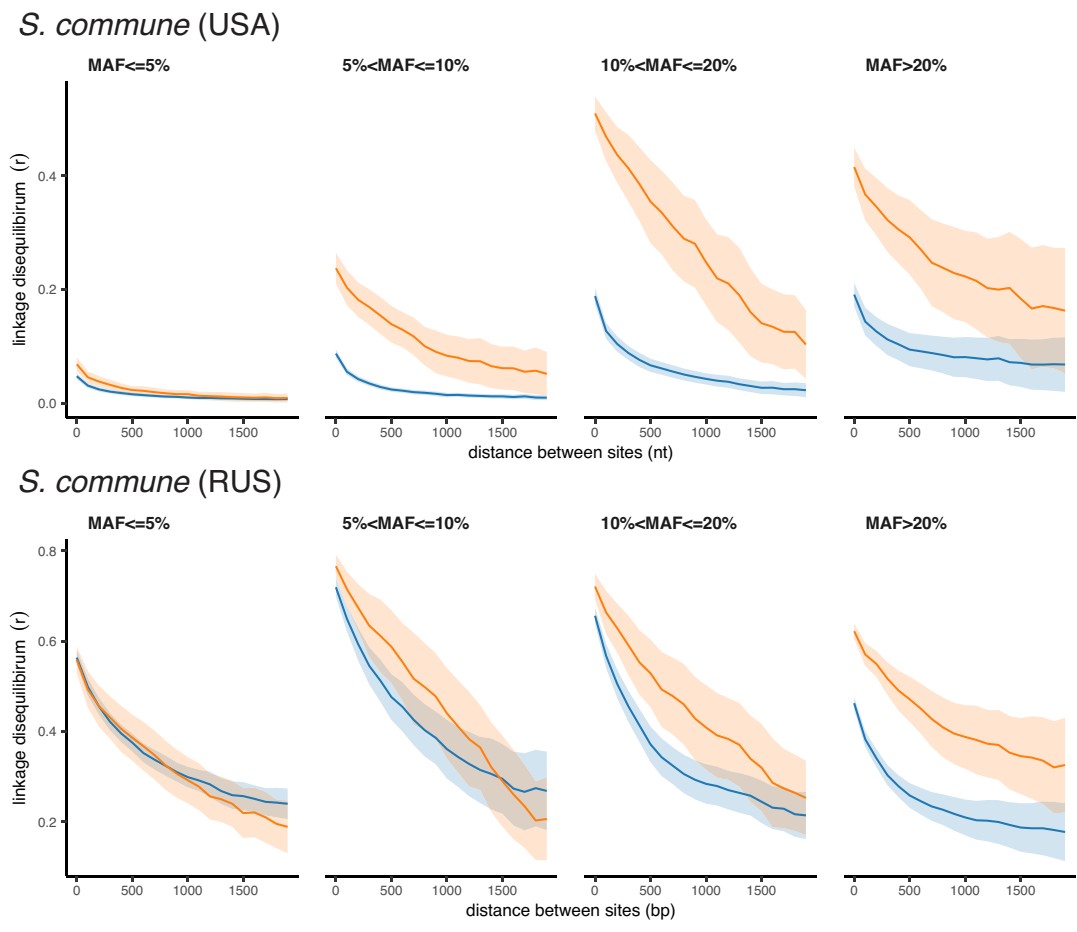

**Appendix 2—figure 1.** Polarized linkage disequilibrium in *S. commune*. LD between nonsynonymous SNPs is shown in orange, and LD between synonymous SNPs is shown in blue. Filled areas indicate SE of LD calculated for each scaffold separately.

### D. melanogaster

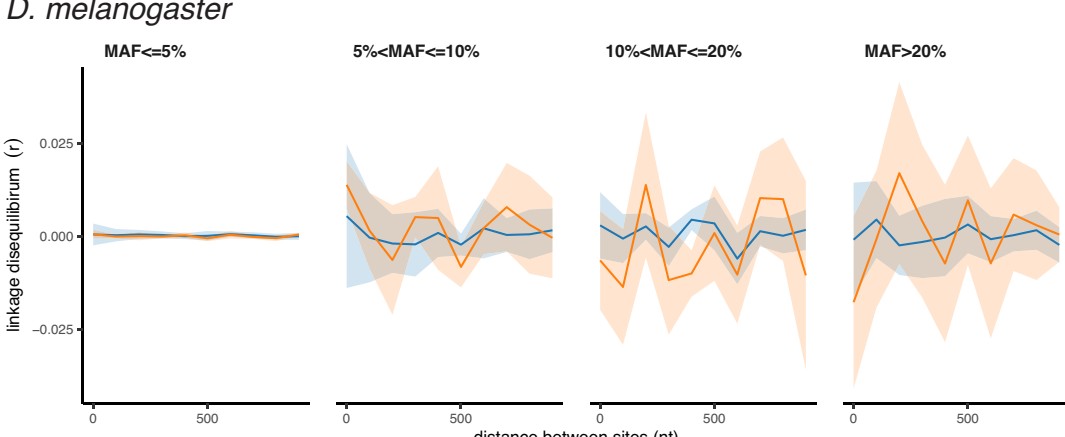

**Appendix 2—figure 2.** Polarized linkage disequilibrium in *D. melanogaster*. LD between nonsynonymous SNPs is shown in orange, and LD between synonymous SNPs is shown in blue. Filled areas indicate SE of LD calculated for each chromosome separately.

### H. sapiens

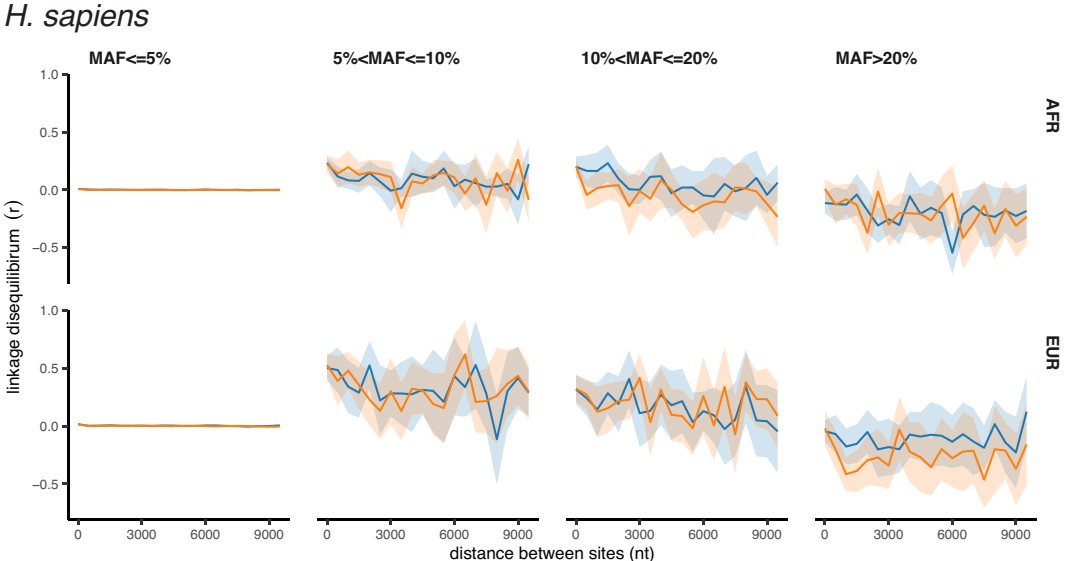

**Appendix 2—figure 3.** Polarized linkage disequilibrium in *H. sapiens*. LD between nonsynonymous SNPs is shown in orange, and LD between synonymous SNPs is shown in blue. Filled areas indicate SE of LD calculated for each chromosome separately.

First, random drift, which affects nearly-neutral synonymous sites more than nonsynonymous sites which are mostly under negative selection, leads to attraction between minor alleles (*Sandler et al., 2021*). Second, negative selection at nonsynonymous sites causes repulsion between rare, deleterious alleles, due to Hill-Robertson interference, even if this selection does not involve any epistasis (*Comeron et al., 2008*; *Garcia and Lohmueller, 2021*; *Hill and Robertson, 1966*; *Appendix 2—figure 4*). Third, there are data on negative epistasis in this selection, which also should lead to repulsion of deleterious alleles and, thus, negative LD between rare nonsynonymous alleles (*Garcia and Lohmueller, 2021*; *Sandler et al., 2021*; *Sohail et al., 2017*). The first and the second factors are weak and can produce noticeable LD only between tightly linked loci, while the third factor may generate even long-range LD. By contrast, $LD_{nonsyn} > LD_{syn}$ can be explained only by positive epistasis in selection at nonsynonymous sites.

Although negative selection generally results in $LD_{nonsyn} < LD_{syn}$, our simulations demonstrated that Hill-Robertson interference without epistasis can produce attraction between minor alleles under a

rather restrictive set of conditions. In these simulations, deleterious polymorphisms can achieve high frequency only in regions of low recombination, leading to $LD_{nonsyn} > LD_{syn}$ for extremely high MAF (*Appendix 2—figure 5A*). However, this effect does not hold if assuming unequal fitness effects of deleterious mutations (*Appendix 2—figure 5B*) or while merging SNPs of different frequencies together (*Appendix 2—figure 5C*).

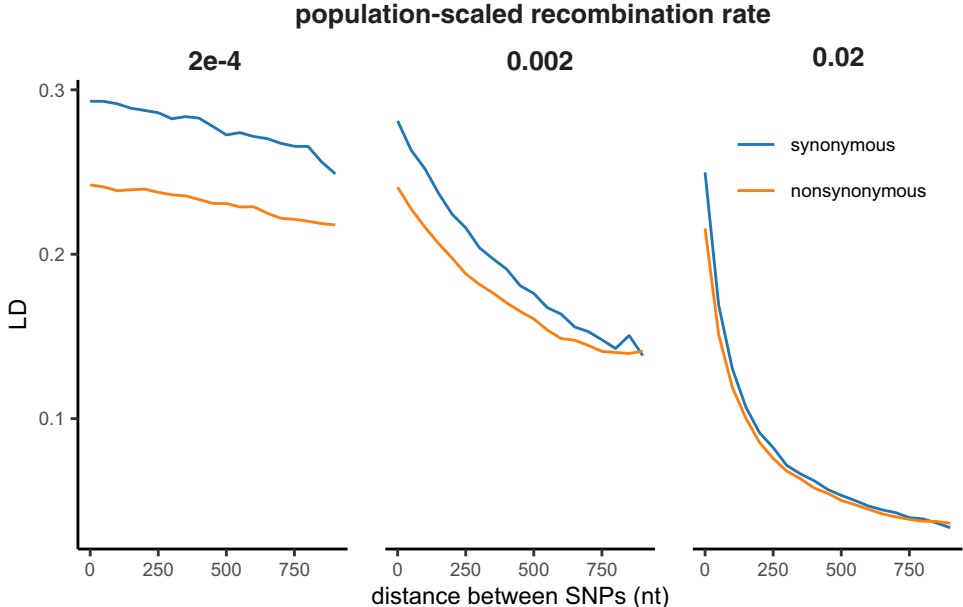

**Appendix 2—figure 4.** $LD_{nonsyn}$ and $LD_{syn}$ in simulations under weak negative selection. LD between synonymous (blue, selection coefficient $N_e s = 0$) and nonsynonymous (orange, $N_e s = -1$) variants under varying recombination rate. Only SNPs with MAF >0.05 are shown. Simulated haploid population size N=2000, sequence length L=1000 bp.

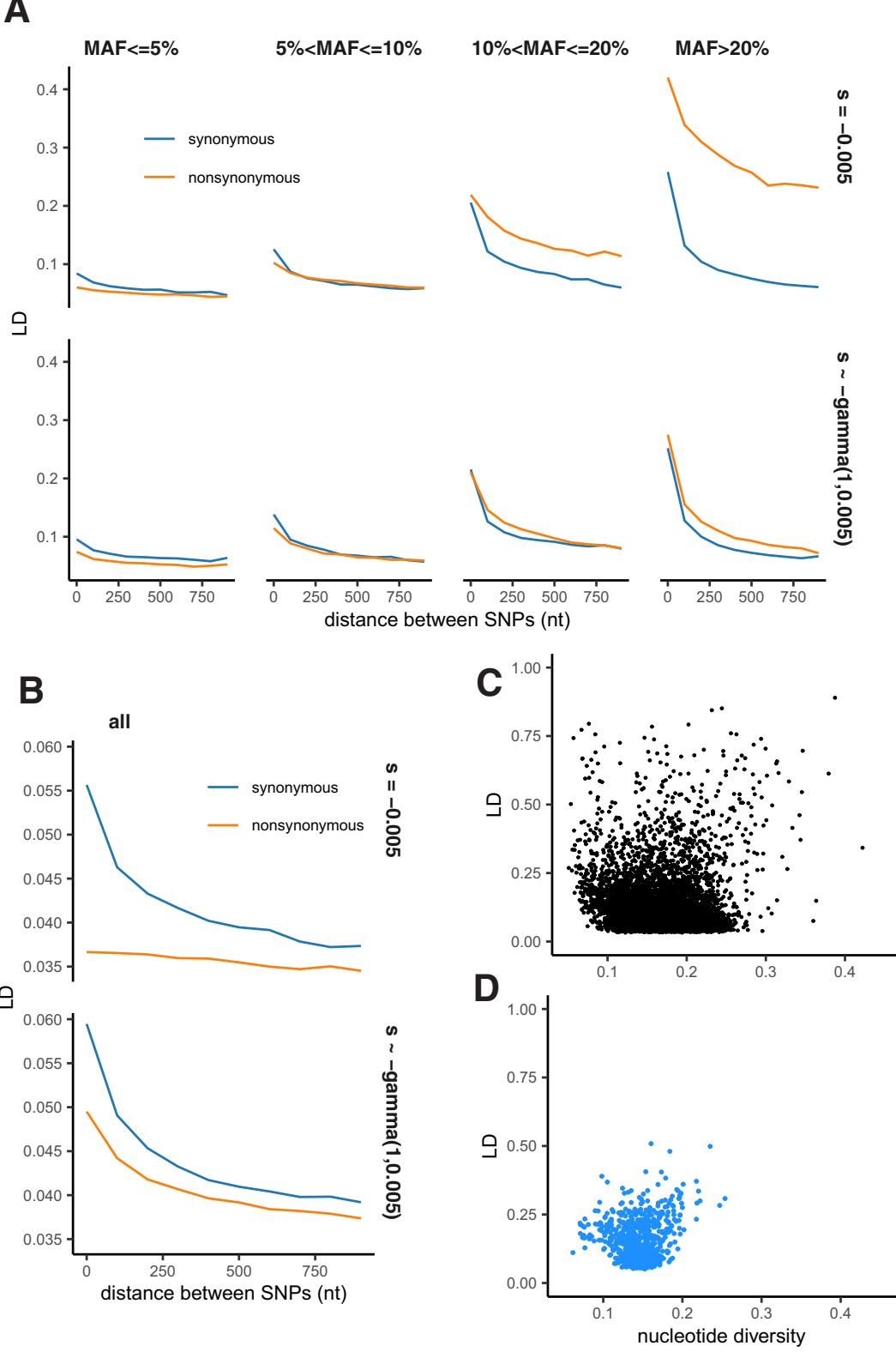

**Appendix 2—figure 5.** Patterns of LD in simulations under Hill-Robertson interference. (**A**) LD between nonsynonymous and synonymous pairs of SNPs split by MAF. (**B**) LD between all pairs of nonsynonymous and synonymous SNPs pooled together. (**A–B**) Simulated haploid population size N=2000, sequence length L=1000 bp. Top panels - selection coefficients of all nonsynonymous mutations are equal to –0.005 ($N_e s$ = –10); bottom panels - *Appendix 2—figure 5 continued on next page*

*Appendix 2—figure 5 continued*
selection coefficients of nonsynonymous mutations are gamma-distributed with parameters rate = 1, scale = 0.005. (**C**) LD and nucleotide diversity within genes of the USA population of *S. commune* (each point represents one gene). (**D**) LD and nucleotide diversity obtained in simulations.

## Appendix 3

### Nucleotide diversity in *S. commune* populations

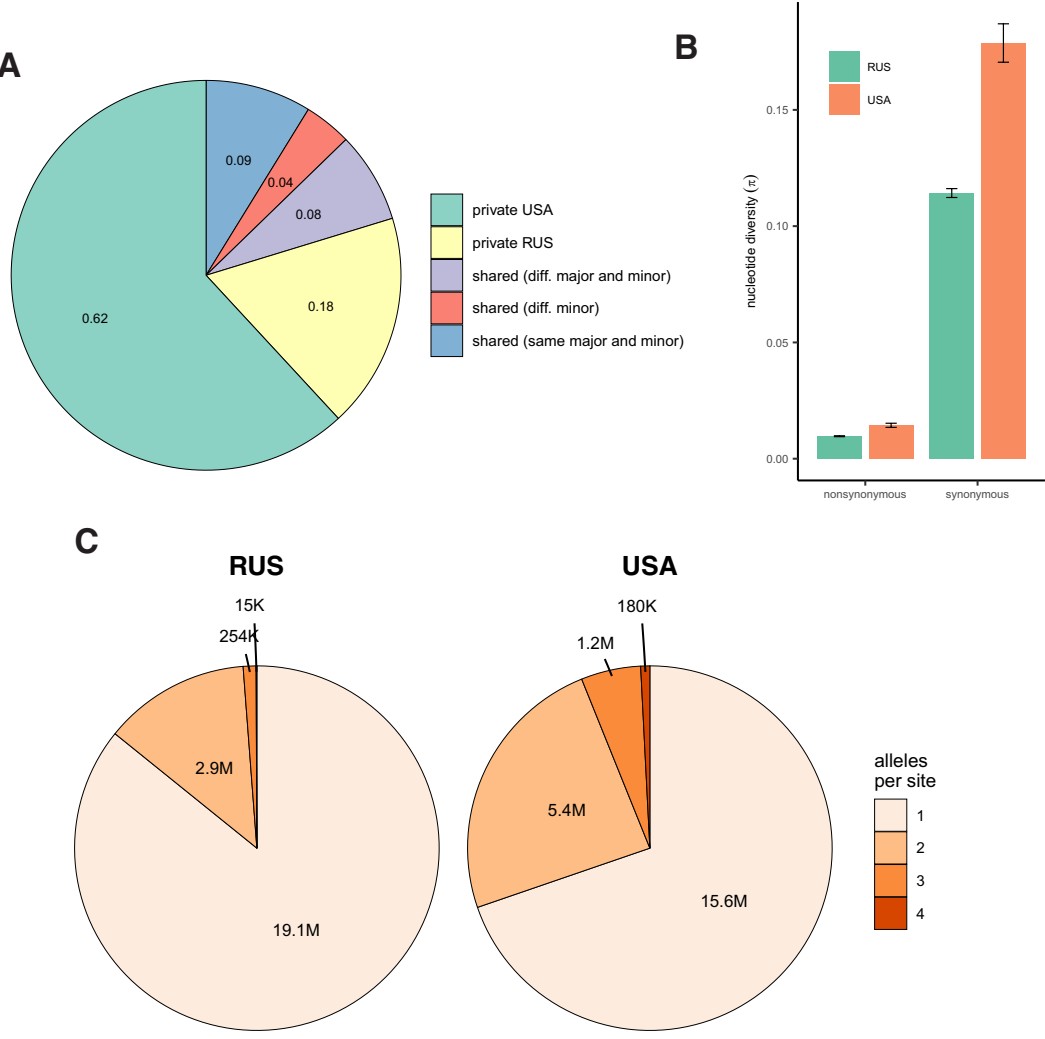

**Appendix 3—figure 1.** Patterns of nucleotide diversity in *S. commune*. (**A**) The fraction of private and shared biallelic SNPs. (**B**) Within-population nucleotide diversity at different classes of sites (measured as π without Jukes-Cantor correction). (**C**) The number of monomorphic and polymorphic sites in the multiple whole-genome alignments of *S. commune* genomes.

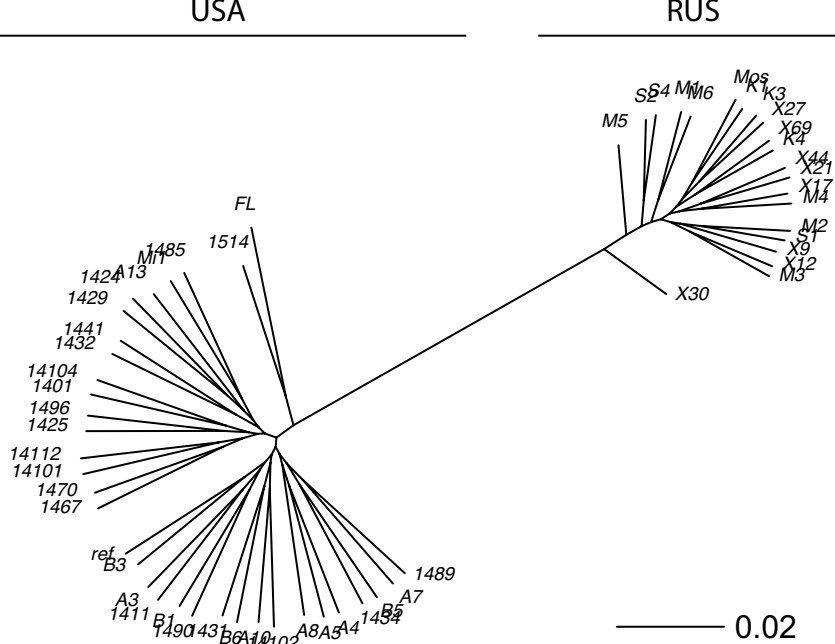

**Appendix 3—figure 2.** The reconstructed phylogeny of *S. commune*. USA and Russian populations of *S. commune* are highly divergent while having almost no within-population structure. Genetic distance is measured in nucleotide differences, the phylogeny is reconstructed based on the multiple whole-genome alignment.

**Appendix 3—table 1.** List of genes with pairs physically adjacent protein sites be under higher LD than pairs of distant sites.

p-values are calculated with chi-square test and adjisted using Benjamini-Hochberg multiple testing correction.

| cog number | 90% $r^2$ quantile | # distant & high LD | # close & high LD | # distant & low LD | # close & low LD | OR | p-value | q-value | aligned PDB ID |
|---|---|---|---|---|---|---|---|---|---|
| RUS population | | | | | | | | | |
| 10,789 | 1.00 | 6 | 6 | 85 | 8 | 10.63 | 4.3E-04 | 8.4E-03 | 4N6Q A |
| 7636 | 1.00 | 40 | 16 | 263 | 11 | 9.56 | 5.2E-09 | 5.5E-07 | 4FQG A |
| 11,223 | 1.00 | 7 | 9 | 87 | 12 | 9.32 | 1.0E-04 | 2.8E-03 | 1S3S G |
| 9853 | 1.00 | 21 | 22 | 261 | 32 | 8.54 | 8.7E-11 | 1.7E-08 | 4 × 00 A |
| 17,085 | 0.19 | 16 | 6 | 183 | 11 | 6.24 | 1.6E-03 | 2.1E-02 | 1NLT A |
| 12,357 | 1.00 | 56 | 30 | 245 | 24 | 5.47 | 1.5E-08 | 1.4E-06 | 2GUY A |
| 1037 | 0.63 | 26 | 13 | 113 | 11 | 5.14 | 4.6E-04 | 8.8E-03 | 1K8F A |
| 6153 | 0.81 | 39 | 14 | 126 | 9 | 5.03 | 5.2E-04 | 9.7E-03 | 5GVH A |
| 14,273 | 0.38 | 22 | 9 | 244 | 21 | 4.75 | 7.5E-04 | 1.3E-02 | 3LCC A |
| 5725 | 0.38 | 26 | 15 | 312 | 38 | 4.74 | 1.6E-05 | 6.3E-04 | 1TA3 B |
| 18,561 | 1.00 | 69 | 32 | 558 | 55 | 4.71 | 3.0E-10 | 4.8E-08 | 1KSG A |
| 3052 | 1.00 | 91 | 25 | 222 | 13 | 4.69 | 1.3E-05 | 5.4E-04 | 4U9V B |
| 3876 | 0.80 | 38 | 22 | 373 | 47 | 4.59 | 4.1E-07 | 3.0E-05 | 1W63 A |
| 4779 | 0.68 | 80 | 26 | 873 | 63 | 4.50 | 1.6E-09 | 2.1E-07 | 2GJL A |
| 16,912 | 1.00 | 172 | 54 | 1383 | 99 | 4.39 | 9.1E-17 | 8.8E-14 | 1WKR A |

*Appendix 3—table 1 Continued on next page*

*Appendix 3—table 1 Continued*

| cog number | 90% $r^2$ quantile | # distant & high LD | # close & high LD | # distant & low LD | # close & low LD | OR | p-value | q-value | aligned PDB ID |
|---|---|---|---|---|---|---|---|---|---|
| 14,670 | 0.82 | 25 | 8 | 273 | 20 | 4.37 | 2.2E-03 | 2.6E-02 | 6C6N A |
| 14,338 | 1.00 | 110 | 28 | 150 | 9 | 4.24 | 2.8E-04 | 6.6E-03 | 6J3E A |
| 8942 | 1.00 | 35 | 10 | 279 | 19 | 4.20 | 1.1E-03 | 1.6E-02 | 3DH1 A |
| 3214 | 1.00 | 37 | 9 | 189 | 11 | 4.18 | 4.4E-03 | 4.2E-02 | 1SZN A |
| 1413 | 1.00 | 78 | 24 | 253 | 19 | 4.10 | 1.8E-05 | 6.8E-04 | 5L3Q B |
| 7650 | 1.00 | 75 | 29 | 201 | 19 | 4.09 | 1.2E-05 | 5.1E-04 | 5EBE B |
| 1071 | 1.00 | 178 | 54 | 1002 | 75 | 4.05 | 1.0E-13 | 2.4E-11 | 1SXJ D |
| 18,096 | 0.64 | 42 | 16 | 462 | 45 | 3.91 | 3.7E-05 | 1.2E-03 | 1WPX A |
| 13,142 | 0.81 | 57 | 9 | 562 | 23 | 3.86 | 1.6E-03 | 2.1E-02 | 5GHE A |
| 16,593 | 0.59 | 118 | 34 | 954 | 72 | 3.82 | 1.7E-09 | 2.1E-07 | 3WDO A |
| 14,325 | 1.00 | 133 | 28 | 621 | 36 | 3.63 | 1.1E-06 | 6.8E-05 | 5U03 A |
| 10,827 | 0.80 | 25 | 11 | 286 | 35 | 3.60 | 2.1E-03 | 2.5E-02 | 2IHO A |
| 10,077 | 0.44 | 38 | 11 | 397 | 32 | 3.59 | 1.3E-03 | 2.0E-02 | 3AKF A |
| 9626 | 0.78 | 47 | 9 | 468 | 25 | 3.58 | 3.2E-03 | 3.4E-02 | 2CVF A |
| 10,648 | 0.75 | 59 | 20 | 587 | 56 | 3.55 | 1.4E-05 | 5.6E-04 | 3I83 A |
| 8,095 | 1.00 | 192 | 60 | 1686 | 151 | 3.49 | 3.2E-14 | 1.0E-11 | 2YMU A |
| 5372 | 0.56 | 88 | 33 | 914 | 99 | 3.46 | 3.3E-08 | 2.9E-06 | 1RGI G |
| 14,269 | 1.00 | 105 | 23 | 426 | 27 | 3.46 | 4.1E-05 | 1.3E-03 | 5UJ8 E |
| 14,404 | 1.00 | 54 | 16 | 374 | 33 | 3.36 | 4.0E-04 | 8.0E-03 | 4IDA A |
| 17,420 | 0.34 | 32 | 12 | 340 | 39 | 3.27 | 2.4E-03 | 2.8E-02 | 1W9P A |
| 6507 | 0.71 | 63 | 13 | 620 | 40 | 3.20 | 9.9E-04 | 1.6E-02 | 1AUA A |
| 9423 | 1.00 | 60 | 11 | 401 | 23 | 3.20 | 4.4E-03 | 4.2E-02 | 4Y42 A |
| 7878 | 0.78 | 41 | 17 | 446 | 58 | 3.19 | 3.5E-04 | 8.0E-03 | 4TYW A |
| 3307 | 1.00 | 87 | 17 | 720 | 45 | 3.13 | 2.3E-04 | 5.8E-03 | 6DVH A |
| 6285 | 0.27 | 55 | 13 | 565 | 43 | 3.11 | 1.4E-03 | 2.1E-02 | 2VWS A |
| 10,049 | 0.83 | 68 | 16 | 668 | 51 | 3.08 | 4.0E-04 | 8.0E-03 | 1KH4 A |
| 6148 | 1.00 | 628 | 131 | 1361 | 93 | 3.05 | 1.6E-15 | 7.7E-13 | 4CHT A |
| 14,511 | 0.63 | 42 | 13 | 414 | 44 | 2.91 | 3.7E-03 | 3.7E-02 | 3HG7 A |
| 2522 | 0.23 | 46 | 16 | 492 | 60 | 2.85 | 1.5E-03 | 2.1E-02 | 5L0R A |
| 5,375 | 0.67 | 45 | 18 | 448 | 63 | 2.84 | 9.6E-04 | 1.5E-02 | 2WZO A |
| 73 | 0.21 | 68 | 20 | 715 | 74 | 2.84 | 2.5E-04 | 6.2E-03 | 1WPX A |
| 12,131 | 1.00 | 210 | 35 | 816 | 48 | 2.83 | 8.7E-06 | 4.0E-04 | 6GKV A |
| 1097 | 1.00 | 63 | 17 | 534 | 51 | 2.83 | 1.1E-03 | 1.6E-02 | 2IW0 A |
| 18,360 | 1.00 | 174 | 35 | 924 | 66 | 2.82 | 3.6E-06 | 2.0E-04 | 1ULT A |
| 5930 | 0.38 | 57 | 16 | 594 | 60 | 2.78 | 1.5E-03 | 2.1E-02 | 2 PXX A |
| 8261 | 0.81 | 112 | 42 | 930 | 129 | 2.70 | 9.4E-07 | 6.5E-05 | 4QNW A |
| 18,092 | 1.00 | 83 | 22 | 794 | 78 | 2.70 | 2.5E-04 | 6.1E-03 | 4QJY A |
| 1060 | 0.78 | 65 | 16 | 533 | 50 | 2.62 | 3.2E-03 | 3.3E-02 | 3WXB A |

*Appendix 3—table 1 Continued on next page*

*Appendix 3—table 1 Continued*

| cog number | 90% r² quantile | # distant & high LD | # close & high LD | # distant & low LD | # close & low LD | OR | p-value | q-value | aligned PDB ID |
|---|---|---|---|---|---|---|---|---|---|
| 17,037 | 0.31 | 95 | 19 | 918 | 70 | 2.62 | 7.4E-04 | 1.3E-02 | 5YHP A |
| 15,353 | 1.00 | 109 | 26 | 944 | 86 | 2.62 | 1.0E-04 | 2.8E-03 | 3WNV A |
| 7784 | 1.00 | 120 | 15 | 929 | 45 | 2.58 | 3.5E-03 | 3.5E-02 | 3L4G B |
| 10,236 | 0.53 | 182 | 35 | 1810 | 135 | 2.58 | 3.5E-06 | 2.0E-04 | 1Q6X A |
| 2011 | 0.34 | 90 | 17 | 889 | 67 | 2.51 | 2.4E-03 | 2.7E-02 | 3A1K A |
| 8572 | 1.00 | 167 | 32 | 976 | 76 | 2.46 | 8.1E-05 | 2.4E-03 | 4AH6 A |
| 14,282 | 0.81 | 153 | 18 | 1251 | 60 | 2.45 | 2.0E-03 | 2.4E-02 | 3QM4 A |
| 7836 | 0.78 | 83 | 17 | 685 | 58 | 2.42 | 4.4E-03 | 4.2E-02 | 1DQW A |
| 3610 | 0.46 | 196 | 45 | 1946 | 185 | 2.42 | 1.2E-06 | 7.3E-05 | 4BKX B |
| 8725 | 0.63 | 71 | 18 | 669 | 71 | 2.39 | 4.0E-03 | 4.0E-02 | 6G6M A |
| 11,096 | 0.48 | 64 | 20 | 657 | 87 | 2.36 | 3.0E-03 | 3.2E-02 | 3AKF A |
| 6520 | 0.75 | 86 | 28 | 599 | 84 | 2.32 | 8.3E-04 | 1.4E-02 | 4K3A A |
| 12,399 | 1.00 | 610 | 107 | 994 | 76 | 2.29 | 1.4E-07 | 1.1E-05 | 1JZQ A |
| 8945 | 1.00 | 147 | 32 | 558 | 53 | 2.29 | 7.9E-04 | 1.3E-02 | 3E5M A |
| 1744 | 1.00 | 148 | 45 | 616 | 83 | 2.26 | 9.7E-05 | 2.8E-03 | 1C7J A |
| 16,360 | 0.82 | 134 | 34 | 1297 | 149 | 2.21 | 2.0E-04 | 5.3E-03 | 3WTC A |
| 12,853 | 0.53 | 168 | 31 | 1634 | 139 | 2.17 | 3.8E-04 | 8.0E-03 | 6H7D A |
| 14,137 | 0.64 | 118 | 27 | 1175 | 127 | 2.12 | 1.7E-03 | 2.1E-02 | 6C5B A |
| 7106 | 0.38 | 124 | 22 | 1167 | 98 | 2.11 | 4.4E-03 | 4.2E-02 | 5K8E A |
| 1523 | 1.00 | 77 | 29 | 402 | 72 | 2.10 | 4.4E-03 | 4.2E-02 | 5Y1B A |
| 2779 | 0.68 | 127 | 27 | 1155 | 120 | 2.05 | 2.8E-03 | 3.0E-02 | 1SXJ B |
| 4275 | 1.00 | 555 | 87 | 957 | 74 | 2.03 | 2.5E-05 | 8.4E-04 | 5VC7 A |
| 17,782 | 1.00 | 184 | 53 | 581 | 83 | 2.02 | 4.1E-04 | 8.0E-03 | 4QNW A |
| 4829 | 1.00 | 350 | 62 | 504 | 46 | 1.94 | 1.7E-03 | 2.1E-02 | 5MXC A |
| 9827 | 0.46 | 237 | 42 | 2273 | 209 | 1.93 | 3.9E-04 | 8.0E-03 | 2VJY A |
| 1520 | 0.45 | 153 | 32 | 1498 | 163 | 1.92 | 2.6E-03 | 2.9E-02 | 3PQV A |
| 4468 | 1.00 | 238 | 48 | 1409 | 148 | 1.92 | 3.6E-04 | 8.0E-03 | 1V9L A |
| 8360 | 1.00 | 852 | 74 | 884 | 40 | 1.92 | 1.5E-03 | 2.1E-02 | 5YLW A |
| 16,987 | 0.53 | 154 | 34 | 1482 | 174 | 1.88 | 2.8E-03 | 3.0E-02 | 3LWT X |
| 935 | 0.65 | 104 | 38 | 1036 | 203 | 1.86 | 3.0E-03 | 3.2E-02 | 3FGA A |
| 11,732 | 0.68 | 118 | 40 | 1076 | 196 | 1.86 | 2.3E-03 | 2.7E-02 | 4C2L A |
| 15,295 | 0.82 | 152 | 41 | 1326 | 193 | 1.85 | 1.7E-03 | 2.1E-02 | 4A69 A |
| 6753 | 1.00 | 246 | 36 | 1868 | 151 | 1.81 | 3.4E-03 | 3.5E-02 | 1SXJ C |
| 13,863 | 1.00 | 319 | 86 | 414 | 62 | 1.80 | 1.6E-03 | 2.1E-02 | 6F43 A |
| USA population | | | | | | | | | |
| 14,970 | 0.53 | 13 | 6 | 160 | 8 | 9.23 | 1.8E-04 | 1.3E-02 | 2VFR A |
| 1536 | 0.65 | 12 | 13 | 184 | 23 | 8.67 | 4.6E-07 | 2.0E-04 | 6AHR E |
| 3618 | 0.20 | 9 | 10 | 139 | 24 | 6.44 | 2.1E-04 | 1.4E-02 | 5LCL B |

*Appendix 3—table 1 Continued on next page*

*Appendix 3—table 1 Continued*

| cog number | 90% r² quantile | # distant & high LD | # close & high LD | # distant & low LD | # close & low LD | OR | p-value | q-value | aligned PDB ID |
|---|---|---|---|---|---|---|---|---|---|
| 18,366 | 0.11 | 44 | 15 | 486 | 41 | 4.04 | 3.5E-05 | 4.5E-03 | 1UPU D |
| 8253 | 0.16 | 44 | 12 | 467 | 35 | 3.64 | 5.8E-04 | 3.4E-02 | 6F87 A |
| 9241 | 0.16 | 56 | 23 | 624 | 81 | 3.16 | 2.6E-05 | 4.2E-03 | 1YCD A |
| 1743 | 0.15 | 49 | 19 | 510 | 64 | 3.09 | 2.1E-04 | 1.4E-02 | 4PEH A |
| 64 | 0.15 | 85 | 27 | 905 | 101 | 2.85 | 1.9E-05 | 3.6E-03 | 2B4Q A |
| 14,128 | 0.11 | 77 | 28 | 804 | 103 | 2.84 | 1.9E-05 | 3.6E-03 | 2 × 8 R A |
| 10,841 | 0.16 | 120 | 20 | 1166 | 69 | 2.82 | 1.5E-04 | 1.2E-02 | 3TIK A |
| 17,174 | 0.64 | 96 | 27 | 679 | 73 | 2.62 | 1.4E-04 | 1.2E-02 | 5EY6 A |
| 5725 | 0.11 | 73 | 30 | 799 | 126 | 2.61 | 5.9E-05 | 6.9E-03 | 1TA3 B |
| 10,834 | 0.37 | 90 | 31 | 936 | 124 | 2.60 | 3.3E-05 | 4.5E-03 | 2QB6 A |
| 1267 | 0.24 | 117 | 29 | 1150 | 116 | 2.46 | 1.0E-04 | 1.0E-02 | 4CPD A |
| 9614 | 0.16 | 149 | 44 | 1550 | 187 | 2.45 | 1.9E-06 | 6.0E-04 | 2VGL B |
| 6148 | 0.53 | 487 | 75 | 4438 | 284 | 2.41 | 1.2E-10 | 1.5E-07 | 4CHT A |
| 161 | 0.10 | 227 | 52 | 2,206 | 210 | 2.41 | 2.0E-07 | 1.3E-04 | 5DNC A |
| 621 | 0.10 | 105 | 35 | 1060 | 152 | 2.32 | 9.1E-05 | 9.7E-03 | 3WG6 A |
| 14,368 | 0.09 | 124 | 26 | 1008 | 92 | 2.30 | 7.4E-04 | 4.1E-02 | 2 × 1 C A |
| 9215 | 0.31 | 161 | 56 | 1546 | 243 | 2.21 | 3.0E-06 | 7.6E-04 | 4QNW A |
| 13,117 | 0.47 | 140 | 44 | 1401 | 226 | 1.95 | 4.5E-04 | 2.8E-02 | 1W9P A |
| 3876 | 0.28 | 232 | 51 | 2244 | 259 | 1.90 | 1.5E-04 | 1.2E-02 | 1W63 A |

**Appendix 3—table 2.** Assembly statistics of 24 genomes of *S. commune* (14 samples from USA population and 7 samples from RUS population).

| sample id | specimen voucher | origin | # contigs | total length (bp) | largest contig (bp) | GC % | N50 | coverage |
|---|---|---|---|---|---|---|---|---|
| 14–01_S62 | WS-M161 | USA; Ann Arbor | 2462 | 37,201,238 | 674,015 | 57.5 | 153,743 | 113.2 |
| 14–101_S73 | WS-M180 | USA; Ann Arbor | 2161 | 36,629,295 | 950,471 | 57.6 | 208,079 | 74.2 |
| 14–102_S74 | WS-M181 | USA; Ann Arbor | 2895 | 37,669,799 | 959,240 | 57.6 | 146,599 | 67.8 |
| 14–104_S75 | WS-M183 | USA; Ann Arbor | 2750 | 37,829,033 | 655,066 | 57.6 | 151,136 | 75.8 |
| 14–112_S77 | WS-M191 | USA; Ann Arbor | 2577 | 37,171,981 | 838,910 | 57.6 | 173,824 | 124.5 |
| 14–11_S63 | WS-M188 | USA; Ann Arbor | 2634 | 37,679,657 | 658,404 | 57.6 | 158,664 | 64.0 |
| 14–25_S64 | WS-M206 | USA; Ann Arbor | 2762 | 38,042,099 | 976,161 | 57.6 | 160,044 | 93.8 |
| 14–29_S65 | WS-M210 | USA; Ann Arbor | 2665 | 37,691,449 | 675,061 | 57.5 | 158,777 | 95.9 |
| 14–31_S66 | WS-M212 | USA; Ann Arbor | 2453 | 37,348,833 | 870,814 | 57.5 | 161,384 | 100.0 |
| 14–32_S67 | WS-M213 | USA; Ann Arbor | 2923 | 37,685,895 | 696,590 | 57.6 | 145,350 | 62.9 |
| 14–34_S68 | WS-M215 | USA; Ann Arbor | 2455 | 37,403,482 | 951,844 | 57.5 | 185,879 | 89.2 |
| 14–67_S84 | WS-M247 | USA; Ann Arbor | 2900 | 37,778,589 | 850,026 | 57.6 | 195,995 | 70.8 |
| 14–70_S69 | WS-M247 | USA; Ann Arbor | 2809 | 37,546,616 | 678,024 | 57.6 | 154,362 | 91.4 |
| 14–85_S70 | WS-M265 | USA; Ann Arbor | 2347 | 37,174,196 | 875,526 | 57.6 | 176,501 | 45.3 |
| 14–89_S71 | WS-M269 | USA; Ann Arbor | 2352 | 37,218,933 | 959,111 | 57.6 | 177,695 | 111.6 |

*Appendix 3—table 2 Continued on next page*

*Appendix 3—table 2 Continued*

| sample id | specimen voucher | origin | # contigs | total length (bp) | largest contig (bp) | GC % | N50 | coverage |
|---|---|---|---|---|---|---|---|---|
| 14–90_S72 | WS-M270 | USA; Ann Arbor | 2957 | 37,322,559 | 739,843 | 57.6 | 139,455 | 110.3 |
| 15–14_S76 | WS-M292 | USA; Florida | 2460 | 37,328,560 | 616,845 | 57.6 | 157,189 | 71.5 |
| X-12_S79 | WS-M12 | Russia; Moscow | 3879 | 38,221,043 | 496,668 | 57.6 | 75,624 | 117.4 |
| X-17_S80 | WS-M18 | Russia; Moscow | 3738 | 37,604,751 | 396,219 | 57.6 | 71,000 | 105.9 |
| X-21_S81 | WS-M22 | Russia; Moscow | 5012 | 39,204,396 | 341,967 | 57.6 | 63,280 | 77.3 |
| X-27_S82 | WS-M28 | Russia; Moscow | 3571 | 37,399,774 | 525,346 | 57.6 | 71,903 | 78.2 |
| X-30_S83 | WS-M31 | Russia; Moscow | 4487 | 38,310,778 | 384,708 | 57.6 | 66,442 | 84.7 |
| X-69_S85 | WS-M70 | Russia; Moscow | 3965 | 38,348,248 | 485,689 | 57.6 | 70,802 | 76.7 |
| X-9_S78 | WS-M9 | Russia; Moscow | 4590 | 38,741,959 | 540,017 | 57.6 | 67,770 | 74.8 |

