## [Editor Report]

This study investigates a highly polymorphic species, the fungus Schizophyllum commune, and finds that, compared to synonymous mutations, levels of linkage disequilibrium between nonsynonymous mutations are higher within genes than between genes. The authors propose this observation may be explained by compensatory interactions between nonsynonymous alleles, pointing to the presence of positive epistasis. These exciting results provide insights into what levels of polymorphism can lead to the emergence of positive epistasis. This paper should be of interest to population geneticists and evolutionary biologists studying the role of natural selection.

---

## [Decision Letter]

**Decision letter after peer review:**

Thank you for submitting your article "Complex fitness landscape shapes variation in a hyperpolymorphic species" for consideration by *eLife*. Your article has been reviewed by 2 peer reviewers, and the evaluation has been overseen by a Reviewing Editor and Detlef Weigel as the Senior Editor. The reviewers have opted to remain anonymous.

Essential revisions:

The reviewers agree that the findings presented in this paper interesting. There were a number of requests for clarifications and analyses that would strengthen the paper. Please find a detailed list of comments and suggestions from each reviewer below.

*Reviewer #1 (Recommendations for the authors):*

1) LD needs to be better defined in the introduction for non-experts. LD in general is a poor way to say that pairs of alleles are found together more or less often than expected.

2) The fact that haplospores can be cultured to sequence haploid DNA needs to be clearly stated in the introduction, as it will dissipate a lot of doubts early on from readers like me who wonder from the start if not all the called variants must be false positives, because how can you map anything properly if heterozygosity is so high? I was by the way pleasantly surprised to find out that the authors used Macse, which is the right choice of aligner but unfortunately not well known and used enough by evolutionary biologists. Was it Macse v1 or v2?

3) Page 5 line 1, about the indistinguishable LD for nonsyn and syn in humans. As stated it sounds in contradiction with Garcia and Lohmueller Plos Genetics 2021. In fact, it is not because the authors focus on MAF>-0.05. But this needs to be better detailed that for that reason there is no contradiction.

4) Page 5 line 3. The authors write about rare alleles but have excluded MAF<0.05? To me, rare alleles is MAF<0.05, so this is a bit confusing to read. The authors need to clarify what they mean by rare alleles here (see my public review main comments). The authors need to better explain why they exclude MAF<0.05. I guess from reading the manuscript that it has to do with (i) the depth of sampling and (ii) negative epistasis between deleterious variants being more prevalent and having the opposite effect and thus potentially masking the positive epistasis signature at lower frequencies? This needs to be much more detailed in the manuscript. About negative epistasis, does it indeed explain why the authors exclude MAF<0.05? In the discussion, I would like to see the authors detailing more if seeing a lot of positive epistasis also implies that there must be abundant negative epistasis between deleterious variants. Is the lack of focus on negative epistasis due to the fact that a proper analysis would require much deeper sampling? These are things I want to know.

5) P3 L19: if the reader misses the first comma in the sentence it takes a completely different meaning. You may want to rephrase.

6) When the authors start mentioning protein structures, I think they should also provide information to the readers in the Results about the nature of protein structure data used (name of the database, number of structures etc.) and not just in the Methods. I like to know about the size of the datasets used in the Results. The authors should also specify in their manuscript that the results with protein structures address potential limitations of the previous nonsyn vs. syn comparison. Clearly this first result cannot be due only to a difference in SFS between nonsyn and syn given the protein structure results.

7) Page 8 line 19. Why only the American and not the Russian population? This needs to be explained.

8) The positive correlation with pN/pS needs to be better explained. Is it because it is more likely then for more positively epistatic variants to find each other? Is it due to lower pN/pS also being correlated to stronger purifying selection/lower frequencies below MAF<0.05?

9) About the proposition of negative frequency-dependent selection, I do think that it explains a lot of the properties of the system. I think however that the authors need to explain better what they mean by that. Is it just about the rare+common allele combinations being deleterious, and too many of those occurring if the rare+rare allele chromosomes reached more intermediate frequencies? Or do the authors think it is more than this and more complicated?

10) A lot of my worries were lifted when I read in the Methods that the authors excluded variants less than 5 amino acids from each other. Up to that point when I was reading, I wondered about how complex multinucleotide mutations could affect the results, and also how the closest nonsynonymous variants may potentially be a bit closer to each other than the closest synonymous variants due to the nature of codons. All these concerns are not valid given the minimal distance of five amino acids, so the authors need to mention it much earlier in the results, together with why they do it.

---

## [Author Response]

Reviewer #1 (Recommendations for the authors):1) LD needs to be better defined in the introduction for non-experts. LD in general is a poor way to say that pairs of alleles are found together more or less often than expected.

We now elaborate on how epistasis may affect linkage disequilibrium between segregating polymorphisms within populations in the Introduction section.

2) The fact that haplospores can be cultured to sequence haploid DNA needs to be clearly stated in the introduction, as it will dissipate a lot of doubts early on from readers like me who wonder from the start if not all the called variants must be false positives, because how can you map anything properly if heterozygosity is so high? I was by the way pleasantly surprised to find out that the authors used Macse, which is the right choice of aligner but unfortunately not well known and used enough by evolutionary biologists. Was it Macse v1 or v2?

We now mention that we sequence haploid DNA in the beginning of the Results section. In this paper, we used macse 2.

3) Page 5 line 1, about the indistinguishable LD for nonsyn and syn in humans. As stated it sounds in contradiction with Garcia and Lohmueller Plos Genetics 2021. In fact, it is not because the authors focus on MAF>-0.05. But this needs to be better detailed that for that reason there is no contradiction.

Our results don’t contradict those of (Garcia and Lohmueller 2021): if we include SNPs with MAF < 0.05 in our analysis, LDnonsyn is lower than LDsyn in both *H. sapiens* and *D. melanogaster* populations, in line with their results. This can be due to one or more of the following factors: Hill-Robertson interference between nonsynonymous variants, higher efficiency of negative epistasis acting on low-frequency variants, and/or lower population frequencies of alleles with MAF < 0.05 in *H. sapiens* and *D. melanogaster,* compared to *S. commune,* because of deeper sampling in these populations (1296 genotypes of *H. sapiens* and 197 genotypes of *D. melanogaster,* compared to 21 and 34 genotypes of *S. commune*). We now add the discussion on LD for different minor allele frequencies and the possible causes of low LDnonsyn between SNPs with low MAF to the main text.

4) Page 5 line 3. The authors write about rare alleles but have excluded MAF<0.05? To me, rare alleles is MAF<0.05, so this is a bit confusing to read. The authors need to clarify what they mean by rare alleles here (see my public review main comments). The authors need to better explain why they exclude MAF<0.05. I guess from reading the manuscript that it has to do with (i) the depth of sampling and (ii) negative epistasis between deleterious variants being more prevalent and having the opposite effect and thus potentially masking the positive epistasis signature at lower frequencies? This needs to be much more detailed in the manuscript. About negative epistasis, does it indeed explain why the authors exclude MAF<0.05? In the discussion, I would like to see the authors detailing more if seeing a lot of positive epistasis also implies that there must be abundant negative epistasis between deleterious variants. Is the lack of focus on negative epistasis due to the fact that a proper analysis would require much deeper sampling? These are things I want to know.

Consistent with the recent studies (Garcia and Lohmueller 2021, Sandler et al., 2021), in the populations of *D. melanogaster* and *H. sapiens,* LDnonsyn is lower than LDsyn for SNPs with MAF < 0.05 (Figure 1 —figure supplement 5 and 6). In *S. commune*, such rare SNPs show LDnonsyn similar to LDsyn (Figure 1 —figure supplement 3). The contradiction between low- and high-frequency polymorphisms can be explained by a stronger effect of Hill-Robertson interference and negative epistasis on alleles with low MAFs.

In this study, we focus on positive epistasis resulting in an excess of LD between nonsynonymous polymorphisms, and provide what we believe is a robust support for it. The issue of potential negative epistasis between them is more complicated. As discussed above, in *S. commune*, negative epistasis can indeed contribute to the absence of strong excess of LDnonsyn between SNPs with low MAF, but this pattern may also be caused by other factors. At the same time, negative epistasis is expected to eliminate combinations of nonsynonymous alleles, decreasing the frequency of deleterious variants and mutational load, which may make negative epistasis between low-frequency polymorphisms hard to detect. Also, for *S. commune* datasets (34 samples from USA population and 21 samples from RUS population), SNPs with MAF < 0.05 correspond to singletons, making such variants more questionable. In order to study negative epistasis within *S. commune* populations, we indeed need to sample more genotypes.

We now include the discussion on LD between SNPs with different minor allele frequencies in the main text.

5) P3 L19: if the reader misses the first comma in the sentence it takes a completely different meaning. You may want to rephrase.

We rephrased the sentence in question.

6) When the authors start mentioning protein structures, I think they should also provide information to the readers in the Results about the nature of protein structure data used (name of the database, number of structures etc.) and not just in the Methods. I like to know about the size of the datasets used in the Results. The authors should also specify in their manuscript that the results with protein structures address potential limitations of the previous nonsyn vs. syn comparison. Clearly this first result cannot be due only to a difference in SFS between nonsyn and syn given the protein structure results.

We add the description of how LD between physically interacting sites was measured to the beginning of the corresponding Results section. We agree that differences in SFS can’t explain the observed excess of LDnonsyn within genes and higher LD between physically interacting sites – we now stress this in the text.

7) Page 8 line 19. Why only the American and not the Russian population? This needs to be explained.

While MAF within haploblocks is significantly higher than in the non-haploblock regions in both American and Russian populations (p-values < 2e-16; Figure 3B, Figure 3 —figure supplement 4), the correlation between the LD within haploblock and MAF is statistically significant only within the American population. For the Russian population, there is no statistically significant correlation (p-value = 0.60). This may be due to the lower genetic diversity and smaller number of samples in the Russian population, resulting in lower statistical power and higher noise. Importantly, our interpretation does not depend on the presence of this correlation.

8) The positive correlation with pN/pS needs to be better explained. Is it because it is more likely then for more positively epistatic variants to find each other? Is it due to lower pN/pS also being correlated to stronger purifying selection/lower frequencies below MAF<0.05?

There might be multiple causes of the positive correlation between the pn/ps and the excess of LDnonsyn, including the efficacy of epistasis being dependent on the allele frequency and local recombination rate. However, it’s hard to conclude whether, for example, epistasis is more efficient for SNPs with higher MAF (so that positively interacting alleles are more likely to co-occur in the same genotype) or, on the contrary, positive epistasis increases the frequency of the deleterious nonsynonymous variants by weakening negative selection acting on them. We now add the discussion on these causes to the main text.

9) About the proposition of negative frequency-dependent selection, I do think that it explains a lot of the properties of the system. I think however that the authors need to explain better what they mean by that. Is it just about the rare+common allele combinations being deleterious, and too many of those occurring if the rare+rare allele chromosomes reached more intermediate frequencies? Or do the authors think it is more than this and more complicated?

In our simulations, we tried to make the NFDS as simple as possible. In these simulations, only one allele is under balancing selection. In epistatic models, the epistatic interactions do not involve this allele, so that balancing selection and epistasis are acting independently. Even in such a model, we were able to reproduce the excess of LDnonsyn, which is created by epistasis and maintained in the whole linked region by balancing selection.

10) A lot of my worries were lifted when I read in the Methods that the authors excluded variants less than 5 amino acids from each other. Up to that point when I was reading, I wondered about how complex multinucleotide mutations could affect the results, and also how the closest nonsynonymous variants may potentially be a bit closer to each other than the closest synonymous variants due to the nature of codons. All these concerns are not valid given the minimal distance of five amino acids, so the authors need to mention it much earlier in the results, together with why they do it.

We now add the corresponding explanations to the beginning of the Results section.